# MR perfusion source mapping depicts venous territories and reveals perfusion modulation during neural activation

Ekin Karasan [1] ✉, Jingjia Chen [1,2,3], Julian Maravilla[1], Zhiyong Zhang[4], Chunlei Liu [1,5] & Michael Lustig[1]

The cerebral venous system plays a crucial role in neurological and vascular conditions, yet its hemodynamics remain underexplored due to its complexity and variability across individuals. To address this, we develop a venous perfusion source mapping method using Displacement Spectrum MRI, a non-contrast technique that leverages blood water as an endogenous tracer. Our technique encodes spatial information into the magnetization of blood water spins during tagging and detects it once the tagged blood reaches the brain's surface, where the signal-to-noise ratio is 3–4 times higher. We resolve the sources of blood entering the imaging slice across short (10 ms) to long (3 s) evolution times, effectively capturing perfusion sources in reverse. This approach enables the measurement of slow venous blood flow, including potential contributions from capillary beds and surrounding tissue. We demonstrate perfusion source mapping in the superior cerebral veins, verify its sensitivity to global perfusion modulation induced by caffeine, and establish its specificity by showing repeatable local perfusion modulation during neural activation. From all blood within the imaging slice, our method localizes the portion originating from an activated region upstream.

The hemodynamics of the cerebral venous system plays a crucial role in ensuring sufficient brain perfusion to maintain cerebral function[1]. While cerebral arterial disorders are well studied and understood, the knowledge of the venous drainage mechanism of the brain is much more limited[2]. This is largely attributed to the person-to-person variability in cerebral venous physiology and the limitations of current imaging technologies[3]. Nevertheless, venous abnormalities play an important role in the pathophysiology of several important neurological conditions. For instance, veins are involved in vascular disorders such as venous sinus thrombosis[4], dural arteriovenous fistulas[5], and idiopathic intracranial hypertension[6]. Emerging evidence suggests potential venous involvement in several other neurological disorders,

including multiple sclerosis, idiopathic Parkinson's disease, leukoaraiosis and normal-pressure hydrocephalus[7,8]. Moreover, it is believed that there is an interplay between venous and Cerebrospinal Fluid (CSF) flows and pressures; however, the exact relationship, particularly concerning venous and CSF hypertension, remains unclear[9].

Venous effects also contribute significantly to functional MRI (fMRI) based on Blood Oxygenation Level Dependent (BOLD) contrast. Veins draining the site of neural activity may carry deoxygenated blood that is displaced from the activated site. This mismatch between locations of BOLD signal and neural activation limits the spatial specificity of the BOLD signal[10–13]. Moreover, veins may exhibit very large BOLD signals, amplifying the SNR near veins. However, this high signal

[1]Department of Electrical Engineering and Computer Sciences, University of California, Berkeley, Berkeley, CA, USA. [2]Bernard and Irene Schwartz Center for Biomedical Imaging, Department of Radiology, New York University Grossman School of Medicine, New York, NY, USA. [3]Center for Advanced Imaging Innovation and Research (CAI2R), Department of Radiology, New York University Grossman School of Medicine, New York, NY, USA. [4]School of Biomedical Engineering, Shanghai Jiao Tong University, Shanghai, China. [5]Helen Wills Neuroscience Institute, University of California, Berkeley, Berkeley, CA, USA. ✉e-mail: ekin_karasan@berkeley.edu

does not necessarily reflect fine-scale neural activity[13–17]. Overall, the venous vasculature has very complex effects on the BOLD signal, and these effects vary across the brain. They are thus difficult to mitigate and cannot be eliminated by averaging out data from an individual subject or across multiple subjects[13]. Noninvasive methods that measure blood perfusion, flow and volume may help decipher the complex biophysics.

Currently, there are several contrast and non-contrast methods that are used to image the blood vessels in the brain, all with major limitations in imaging the venous system. Digital Subtraction Angiography (DSA), albeit being the gold standard to image many vascular disorders involving the venous system, is an invasive procedure that has life-threatening stroke risks[18,19]. Time-resolved, Dynamic Contrast Enhanced (DCE) MRI is an alternative noninvasive tool to study the venous system; however, it is not as reliable as DSA, has limited spatial and temporal resolution and requires exogenous contrast agents[20]. Several noninvasive, non-contrast-agent-based MRI methods have been developed to resolve vascular structure, blood flow velocity and arterial perfusion. Arterial Spin Labeling (ASL) uses blood water as an endogenous tracer by inverting blood magnetization (i.e., labeling) in a slab that crosses the large arterial vessels feeding the brain[21]. The labeled blood perfuses downstream and modulates the signal in images, which can be used to quantify perfusion. While ASL is frequently used to study arterial perfusion, it is not suitable for veins because of the lack of a single labeling plane. Variants such as Velocity-selective ASL (VSASL) can detect venous signals, but often venous signals are considered unwanted and suppressed using techniques like Acceleration-selective ASL (AccASL)[22].

Primary noninvasive tools to probe the venous system are Time of Flight (TOF) MR Venography (MRV), which relies on inflow effects, and phase-contrast (PC) MRI, which encodes instantaneous flow velocity into signal phase[2,23,24]. TOF MRI is limited to imaging the venous structure and provides minimal information on flow dynamics. PC MRI requires very large velocity encoding gradients to capture slower flows, potentially leading to phase offset errors and reducing the accuracy of the measured velocities[25]. Therefore, it is mainly limited to probing flow velocity in sinuses and larger veins. Another technique, Venous Territory Imaging using Remote Sensing (VENTI)[26], is a preliminary method developed to map venous territories that share some similarities and differences with our approach. However, there have been no follow-up developments or validation studies of VENTI.

To address current limitations in venous imaging, we leverage Displacement Spectrum (DiSpect) MRI[27] to develop a non-contrast method for mapping venous perfusion in the brain. The venous system structure is the reverse of the arterial system, namely, capillaries drain into small veins that drain successively into larger and fewer vessels downstream. So we take a unique approach, whose main concepts are illustrated in Fig. 1. The concepts of tagging/labeling and imaging/detection are shown in Fig. 1a. Blood water spins in the entire brain are labeled using DiSpect spatial encoding. The labeled blood starts draining into larger cerebral veins. Eventually, the vessels cross the imaging slice, as depicted in Fig. 1b, where the labeled blood is imaged/detected.

Figure 1c depicts the overall acquisition strategy. Blood within the vessels and the brain tissue itself pulse with each cardiac cycle, but this motion is overall consistent across cardiac cycles. Cardiac triggering is used to synchronize the labeling. Immediately after, the same slice is dynamically imaged as fresh blood enters the imaging slice. Each image captures blood at a longer and longer mixing time relative to the labeling ($t_{\text{mix}_1}, t_{\text{mix}_2}, t_{\text{mix}_3}, \ldots$), during which the signal also experiences $T_1$ decay. The whole acquisition is repeated $N$ times with incrementing DiSpect spatial encodings ($s_1, s_2, \ldots s_N$)[27–29]. The number of repetitions, $N$, is determined based on the desired resolution and field-of-view (FOV) of the perfusion maps. Finally, information from different spatial encodings is combined to obtain the perfusion sources that feed individual vessels within the imaging slice using the Fourier transform.

## Like ASL, but in reverse

A reader may notice the reverse relation to ASL. ASL labels a plane with few major source blood vessels and images their flow into smaller arteries, capillaries and surrounding tissue in a target volume, thus obtaining perfusion maps where labeled blood went to. In our approach, we label a source volume comprising of small veins, capillaries and surrounding tissue that drain into large veins crossing a target plane. We obtain maps of the sources where blood came from— hence capturing venous perfusion sources in reverse.

## Remote detection

One important property of our method is that the labeled blood in the source volume is remotely detected[30,31] at a target plane placed close to the brain's surface, where the signal-to-noise ratio is typically 3–4 times higher due to the proximity to the receiver coils.

## Dynamic imaging

By placing the target imaging slice on large veins where blood is quickly refreshed, the temporal dynamics can be probed without disturbing perfusion upstream. This gives us the ability to resolve perfusion source maps across short (10 ms) to long (3 s) evolution times. Blood sources can be traced regardless of their path and velocity, enabling measurement of slow blood flow in smaller veins and potentially surrounding capillary beds and tissue, as well as fast blood flow in larger veins.

In this work, we demonstrate and validate venous perfusion source mapping with DiSpect in the superior cerebral veins. We verify its sensitivity to detecting global perfusion modulation induced by caffeine. We further establish its specificity by showing that our method can consistently and repeatably detect local perfusion changes due to the neural activation of the motor cortices. Even though imaging is performed at a location downstream of the neural activation, the perfusion source maps can discern the blood within the imaged volume that originates from an activated region upstream (Fig. 1d). Further, to address scan time limitations, we introduce a multi-slice DiSpect experiment, where each slice maps perfusion sources from a smaller region. By combining data from all slices, we map the perfusion dynamics of entire superior veins while significantly reducing acquisition time.

## Results

### Perfusion source mapping acquisition

DiSpect is based on Displacement Encoding with Stimulated Echoes (DENSE)[28,29], a method primarily employed to quantify heart wall displacement and cardiac strain[29,32,33]. However, DENSE is only sensitive to bulk displacements and is unable to discern partial volume displacement effects, for example, those arriving from multiple sources into a single image voxel. DiSpect, a Fourier-encoding variant of DENSE, overcomes this limitation by resolving the multidimensional sources of displaced spins within a voxel. This is achieved through multiple cardiac-gated acquisitions, with gradually increasing DENSE encodings applied during tagging. The pulse sequence of DiSpect is shown in Fig. 2a (see "Methods" section for a more detailed explanation of the acquisition).

At the end of a DiSpect acquisition, we obtain the perfusion source maps for each target voxel in our imaging slice. Importantly, encoding of the target and source dimensions are independent. The target FOV and resolution are determined by the parameters of the image acquisition. In our experiments, we choose to use a single-shot spiral readout to reduce acquisition time, although our method can work with other readouts as well. Conversely, the source FOV and resolution are determined by the step size and maximum amplitude of

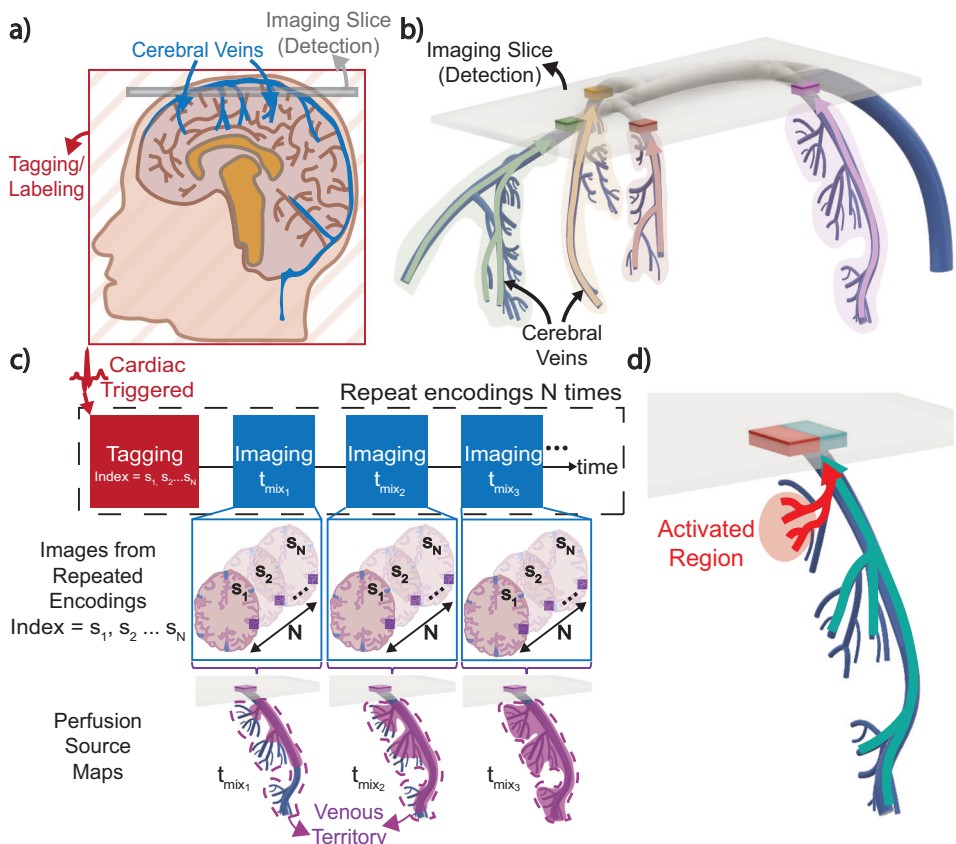

**Fig. 1 | Conceptual framework of venous perfusion source mapping in the superior cerebral veins. a** Schematic of the cerebral system showing the veins that drain into the superior sagittal sinus. During tagging/labeling, spatial information is encoded into the magnetization of blood water spins across the entire brain using Displacement Spectrum (DiSpect) spatial encoding. This labeled blood then drains into larger cerebral veins, eventually crossing the imaging slice where the encoded information is remotely detected. **b** 3D view of the superior cerebral veins and the axial imaging slice. The labeled blood draining through different superior veins enters distinct regions of the imaging slice, allowing the drainage of each vein to be traced independently. **c** The overall acquisition strategy. Cardiac triggering synchronizes the labeling with the cardiac cycle. After labeling, the same slice is imaged dynamically to capture the inflow of fresh blood over increasing mixing times $(t_{mix_1}, t_{mix_2}, t_{mix_3}, \ldots)$. This process is repeated with incrementing displacement encodings $(s_1, s_2, \ldots s_N)$, and the information from all encodings is combined to obtain dynamic perfusion source maps via a Fourier transform. The number of repetitions, $N$, depends on the desired resolution and FOV of the source maps. **d** During neural activation, venous perfusion in the region of activation will be modulated, which can be detected at an imaging slice placed upstream.

the DENSE encodings. Each acquisition with a set of DENSE encoding samples a single point in the Fourier space of the perfusion source maps. Multiple acquisitions with varying DENSE encodings are made to achieve the prescribed source FOV and resolution.

While it is generally possible to resolve the sources of each target voxel in 3D, this process is time-consuming due to the need for numerous acquisitions to fully sample the Fourier space of the perfusion source maps. In our venous imaging experiments, we perform displacement encoding along the left-right (LR) and superior-inferior (SI) directions to create 2D coronal perfusion source maps, projected along the anterior-posterior (AP) direction. By selecting regions from the imaging slice that contain distinct cortical veins, the drainage of each vein can be isolated and traced independently (Fig. 1b). This allows the sources to be projected over a thin slab that is constrained by venous anatomy. By combining information from the perfusion source maps across different evolution times, we can determine the venous territory of each cortical vein.

### Experimental demonstration of venous perfusion source mapping

We begin by validating the ability of DiSpect to assess venous perfusion in the superior cerebral veins. The experimental setup and results are illustrated in Fig. 3. The prescribed imaging slice goes through the superior sagittal sinus and several smaller cortical veins (Fig. 3a). The

target image resolution was $4 \times 4$ mm$^2$, and the in-plane perfusion source map resolution was $6 \times 6$ mm$^2$ (projected over AP). Data was acquired at mixing times between 100 ms and 3 s, in 150 ms increments. Quantitative Susceptibility Mapping (QSM)[34,35] was used to produce a high-spatial-resolution ($0.8 \times 0.8 \times 0.8$ mm$^3$) cerebral venogram. The obtained susceptibility maps were averaged across echo times and the coronal vein structure was visualized using a maximum intensity projection along the AP direction.

Different vessels that flow through the imaging slice can be identified by examining the perfusion source map energy of target image voxels and detecting peaks. Among these vessels, we select four target ROIs from the imaging slice (Fig. 3b), each containing a different superior vein (Fig. 3c). The perfusion source maps for each vein are overlaid on the QSM-venogram of the coronal section containing that vein (Fig. 3d). These maps delineate the sources of blood entering each ROI over 700, 1300 and 1900 ms. The venous territory drained by each vein is marked with a dashed contour (Fig. 3d), determined by the maximum extent of the sources across all mixing times.

The signal intensity in the perfusion source maps reflects the relative proportion of blood from each source position that has displaced into the target imaging voxel over a mixing time period. For example, in Fig. 3b, the yellow perfusion source maps represent the proportion of blood that has displaced into the target voxel shown with a yellow box. While outside the scope of this work, we expect that

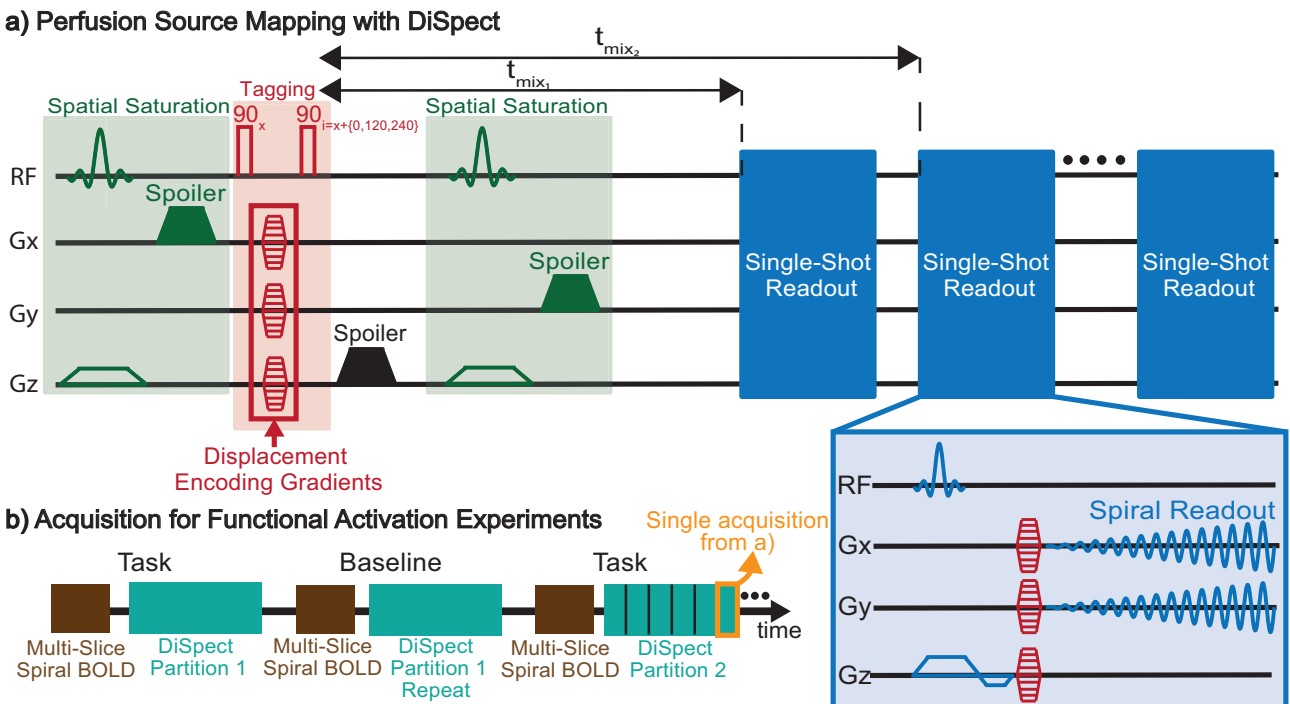

**Fig. 2 | The displacement spectrum (DiSpect) pulse sequence. a** The sequence begins with a tagging module, which cosine modulates the longitudinal magnetization according to the current position of spins. The phase of the second radio-frequency (RF) pulse during tagging is varied in 120° increments in order to perform echo separation (see "Methods"). The displacement encoding gradients are varied in phase-encoding increments to make multiple acquisitions with incrementing displacement encodings. Spatial saturation pulses are placed before and after tagging to suppress unwanted signals. In our experiments, the spatial saturation band was positioned around the imaging slice to suppress signals from static tissue. Images are acquired dynamically after tagging with a single-shot spiral readout, corresponding to different mixing times. The first two mixing times, denoted as $t_{mix_1}$ and $t_{mix_2}$, are marked on the figure. The Gx, Gy, Gz labels denote the x, y, and z gradient waveforms, respectively. **b** Acquisition strategy for the functional activation experiments. The DiSpect acquisition is split into 20 s partitions, each containing 6 repetitions of acquisition shown in (**a**). Each partition is repeated during the task and at baseline before moving to the next. A multi-slice Spiral Blood Oxygenation Level Dependent (BOLD) acquisition was performed between partitions to ensure that activation occurs consistently throughout the scan. Volunteers are instructed to begin and stop the task using automated voice commands.

in order to obtain absolute quantitative maps, a calibration scan will be needed. This calibration scan will estimate the absolute volume of blood displacing into the image voxel, from which quantitative source maps could be computed (see "Methods" section for more details on quantification).

### Ability to detect remote sources in the deep cerebral veins

With cortical veins, the detected perfusion sources mainly originate from cortical regions, since these veins drain cortical areas. To further demonstrate the ability of DiSpect to display remote sources, we performed an additional experiment targeting the deep cerebral veins. Unlike the remainder of our experiments, we performed displacement encoding along the AP and SI directions to obtain 2D sagittal perfusion source maps. The target image resolution was $4 \times 4$ mm$^2$, and the in-plane perfusion source map resolution was $6 \times 6$ mm$^2$ (projected over LR). Data was acquired at mixing times between 100 ms and 3 s, in 150 ms increments.

In this experiment, the imaging slice was placed lower in the brain (Supplementary Fig. 1a), and an ROI was selected on the straight sinus (Supplementary Fig. 1b). The measured perfusion source maps were overlaid on the QSM-venogram of the sagittal section containing the deep cerebral veins. These maps show the sources of blood entering each ROI over 250, 700 and 1150 ms. The venous territory drained by the straight sinus is marked with a dashed contour (Supplementary Fig. 1d), determined by the maximum extent of the sources across all mixing times. The perfusion source maps reveal signal originating from deep cerebral structures at the center of the brain, with sources detected up to 15 cm away from the selected ROI.

### Sensitivity to caffeine-induced global perfusion changes

Caffeine, a vasoactive agent, has been shown to reduce flow velocities in major cerebral arteries[36] and increase venous transit times[37]. We use caffeine to investigate the capability of our method to detect global venous perfusion changes. The subject abstained from caffeine consumption for 72 h prior to the scan. An initial DiSpect acquisition was performed with a target image resolution of $4 \times 4$ mm$^2$ and in-plane perfusion source map resolution of $6 \times 6$ mm$^2$. Following this, the subject orally consumed 225 mg of caffeine, while remaining still in the scanner bed. A second acquisition was performed after a 20-min delay with identical acquisition parameters. This provided two perfusion source maps for each image voxel: one at baseline and one post-caffeine ingestion.

To visualize alterations in venous perfusion, we selected two target vein ROIs from the imaging slice (Fig. 4a, b). First, we averaged the perfusion source maps acquired at baseline and post-caffeine and displayed them at mixing times of 850 and 1600 ms (Fig. 4c). Subsequently, we selected source ROIs within the perfusion source maps of each image voxel and plotted their normalized signal amplitude at different mixing times (Fig. 4d). Our results clearly demonstrated that, in all selected source ROIs, the peak of the time course at baseline occurs earlier than the peak post-caffeine, indicating a delayed venous arrival of blood after caffeine consumption.

We then quantified the post-caffeine delay in blood arrival times at each source location within the perfusion maps. For each source location, we identified the shortest mixing time at which the signal amplitude exceeds 50% of its maximum value across all mixing times. We then calculated the difference between the mixing times identified

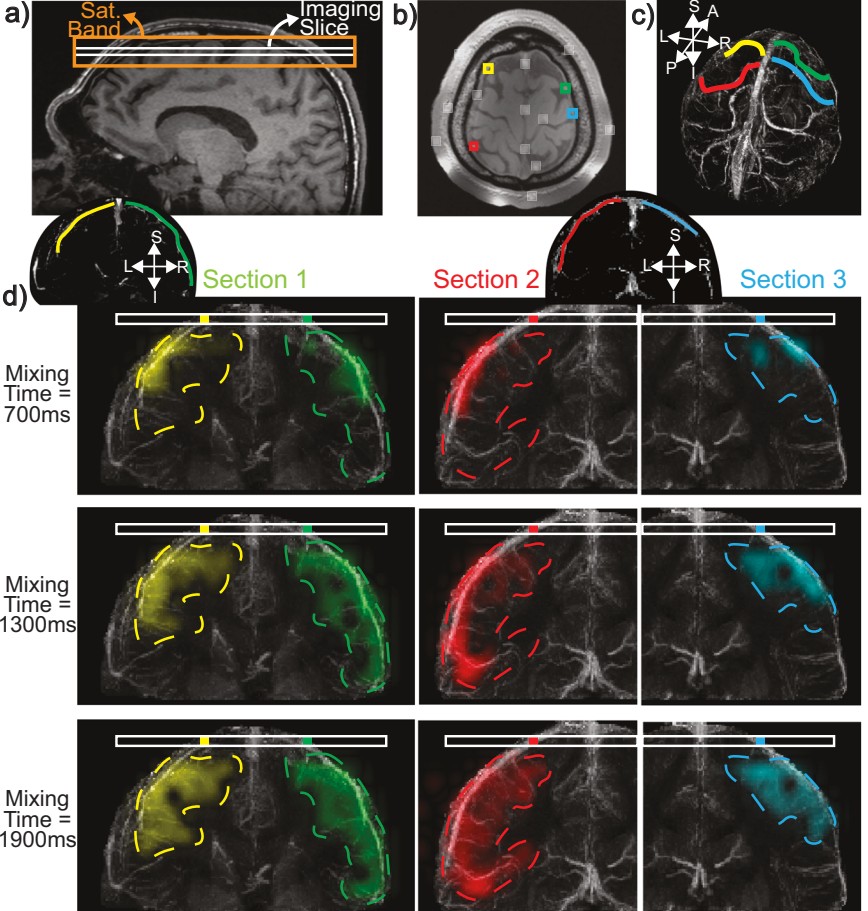

**Fig. 3 | Setup and results for perfusion source mapping of individual superior veins. a** A sagittal image showing the placement of the axial imaging slice and saturation band. **b** From the dataset, different vessels that flow through the imaging slice can be identified by examining the perfusion source map energy of target image voxels and detecting peaks. The detected peaks are overlaid with white ROIs on the image. Four color-coded ROIs are selected out of these regions, each containing a different superior vein that drains into the slice. **c** The color-coded draining veins overlaid on a 3D Quantitative Susceptibility Mapping (QSM) venogram. Anatomical orientations for the QSM-venogram are labeled as follows: Anterior (A), Posterior (P), Superior (S), Inferior (I), Left (L), and Right (R). **d** 2D perfusion maps from each ROI, overlaid on the coronal sections of QSM-venograms, showing the ability to trace the sources of blood. Over short mixing times, blood remains in the large vein. For longer times, the diffused maps indicate blood coming from smaller veins and potentially from capillary beds and surrounding tissue. The coronal projection of the venous territory, obtained from the perfusion source maps, is shown with dashed contours for each vein. Source data are provided in ref. 54.

at baseline and post-caffeine, providing the delay in arrival time. The obtained delay maps are plotted over a range of 0–400 ms (Fig. 4e).

## Specificity to local perfusion modulation during neural activation

Unlike caffeine, which influences venous perfusion globally, neural activation induces local modulations of venous perfusion. A modified DiSpect pulse sequence was developed to investigate these perfusion changes (Fig. 2b). To maintain short functional task durations, the full DiSpect acquisition was split into 20-s partitions. At the beginning of each partition, the subjects were prompted with an automated voice command to "begin" or "stop" a motor cortex functional task. Each partition was repeated during the task and at baseline before proceeding to the next. A multi-slice Spiral-BOLD (TR = 2 s, 5 TRs per partition) acquisition was performed at the beginning of each DiSpect partition to verify consistent neural activation throughout the scan. DiSpect data from different partitions was combined to obtain two perfusion source maps: one at baseline and one during task.

Two subjects (Subjects 1 and 2) were scanned with the modified DiSpect acquisition, each undergoing two scans on two separate days. For the first scan of Subject 1, an in-plane perfusion source map resolution of $8 \times 8$ mm$^2$ was used while the subject was instructed to

squeeze both hands during the task. To achieve a more detailed visualization of the right motor cortex, a second scan was performed with a $6 \times 6$ mm$^2$ in-plane perfusion source map resolution, while the subject was instructed to solely squeeze their left hand. For both scans of Subject 2, the same acquisition with an $8 \times 8$ mm$^2$ in-plane perfusion source map resolution was performed, with the subject instructed to squeeze only their right hand. In addition, a single scan was conducted on a third subject (Subject 3), with an $8 \times 8$ mm$^2$ in-plane perfusion source map resolution, during which the subject was instructed to squeeze their left hand.

At the end of each scan session, a standard 2D EPI-BOLD acquisition was conducted, where the subject was instructed every 20 s to alternate between performing the task and resting, for an overall duration of 5 min. The data was used to obtain reference BOLD activation maps.

To illustrate local perfusion modulation, we selected four veins from Subject 1 during bilateral motor cortex activation: two (orange) veins draining the activated motor cortices and two (blue) veins selected as controls (Fig. 5a, b). We display the perfusion source maps and the percent change between baseline and task perfusion source maps at mixing times of 1000 and 1750 ms (Fig. 5d). In the percent change maps, any region exhibiting positive percentage

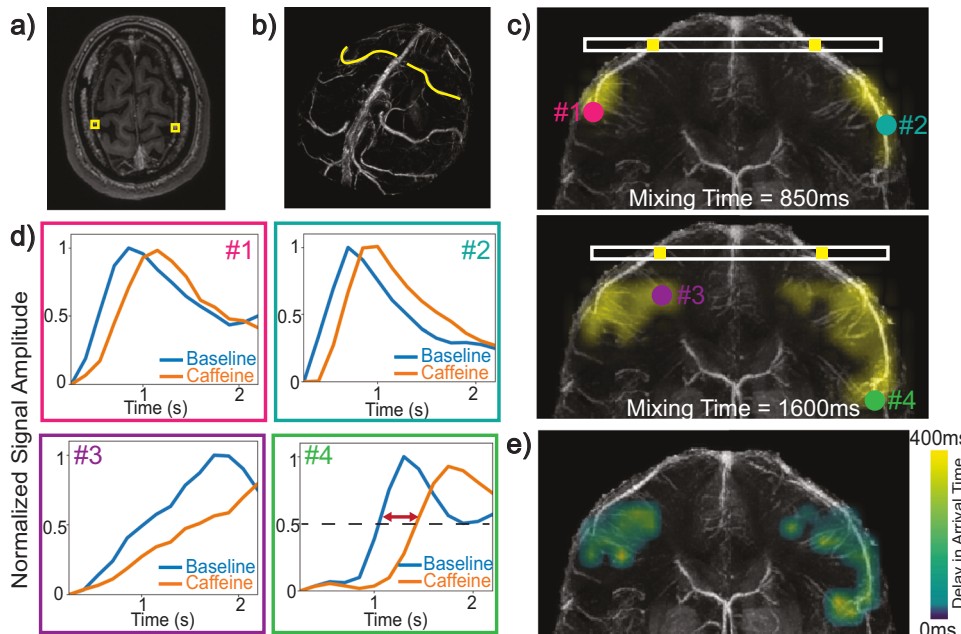

**Fig. 4 | Experimental results verifying sensitivity of perfusion source mapping to caffeine-induced global perfusion changes. a** Imaged slice with two selected vein ROIs. **b** Selected veins overlaid on a 3D venogram. **c** The perfusion source maps of the selected veins at two mixing times (850 ms and 1600 ms). **d** Two color-coded source ROIs are selected from the perfusion source maps of each vein. A circle corresponding to each ROI is shown in its respective color in (**c**). Pink and purple ROIs (ROIs 1 and 3) are selected from the vein on the left, whereas the blue and green ROIs (ROIs 2 and 4) are from the vein on the right. The signal amplitude of each ROI is normalized, corrected for $T_1$ decay and plotted at different mixing times for baseline and post-caffeine states. For all source ROIs, the peak of the baseline signal occurs earlier than post-caffeine, indicating a delayed arrival after caffeine ingestion. **e** The delay in arrival time was estimated for each source location in the perfusion source maps and displayed for a range of 0–400 ms. Source data are provided in ref. 54.

change, where the task signal exceeds baseline, is displayed in red, whereas any region with negative percentage change is displayed in blue. Coronal projections of the BOLD T-scores for the sections containing each vein are shown in the bottom row of Fig. 5d. The venous territory that each vein is draining is outlined with a dashed contour. Similarly, the results from the first scan of Subject 2 with left motor cortex activation are displayed in Fig. 6. Here, one activated vein (orange) draining the left motor cortex and one control vein (blue) were selected (Fig. 6a, b). Perfusion source maps and their percentage change during task relative to the baseline measurement at mixing times of 550 and 1150 ms are displayed in Fig. 6d. Additionally, the results from Subject 3 are shown in Supplementary Fig. 2. For this subject, one activated vein (orange) draining the right motor cortex and one control vein (blue) were selected (Supplementary Fig. 2a, b). Perfusion source maps and their percentage change during task relative to baseline at mixing times of 400 and 1150 ms are displayed in Supplementary Fig. 2d. The displayed mixing times for each subject were selected to highlight the most significant percentage changes.

In all subjects, we observed a significant local increase in the perfusion source map signal during the task, particularly near veins draining the activated motor cortices. This indicates that an increased amount of blood is arriving from those source locations into the target image voxel across the corresponding mixing time. Notably, this increase was observed in the regions of the venous territory close to the BOLD activation, while other regions of the territory exhibited minimal or no modulation. Subject 1 also exhibited a significant local decrease in signal during task in the right motor cortex, located slightly further away from the BOLD activated region. The perfusion source maps of the control veins in all subjects showed little or no percent change between baseline and task.

To illustrate repeatability, we show the percentage change in perfusion source maps for the vein draining the same activated motor cortex from two scans of Subjects 1 and 2 (Fig. 7). For Subject 1, we chose the vein draining the right motor cortex (Fig. 7a), while for Subject 2, we selected the vein draining the left motor cortex (Fig. 7b).

To assess the consistency of observed modulations, T-tests were conducted on the initial scan data from Subjects 1 and 2, comparing the mixing time signal series at baseline and during task at every source location in the perfusion source maps. The T-statistic images with a significance threshold of $p = 0.05$ are displayed in Fig. 8a, highlighting regions of the venous territory showing the largest perfusion modulation. For Subject 1 with bilateral motor cortex activation, both veins drain activated motor cortices, whereas for Subject 2 with unilateral activation, only one vein drains the activated left motor cortex.

Next, ROIs near the statistically significant regions identified by the T-test were selected from the perfusion source maps (Fig. 8a). For each ROI, the source signal amplitude was displayed at baseline and during task (Fig. 8b). This analysis illustrates how the source signal in these regions evolves over mixing time during the two states. The signal during task exceeds baseline for all regions displaying positive T-values. In contrast, for ROI #2 located near the area showing a negative T-value, the baseline signal clearly exceeded the signal during task.

In order to demonstrate that the motor cortices are getting consistently activated across the DiSpect partitions, we analyzed the data collected during the Spiral-BOLD acquisitions performed at the beginning of each partition. We selected three ROIs within the functionally activated regions (Fig. 8c). Next, we display the signal from the custom Spiral-BOLD acquisitions for each selected ROI (Fig. 8d). The red markers indicate the last Spiral-BOLD acquisition during task, and the green markers indicate the last acquisition during rest. The plotted signal amplitudes indicate that all ROIs exhibit consistent activation throughout the scan, as anticipated.

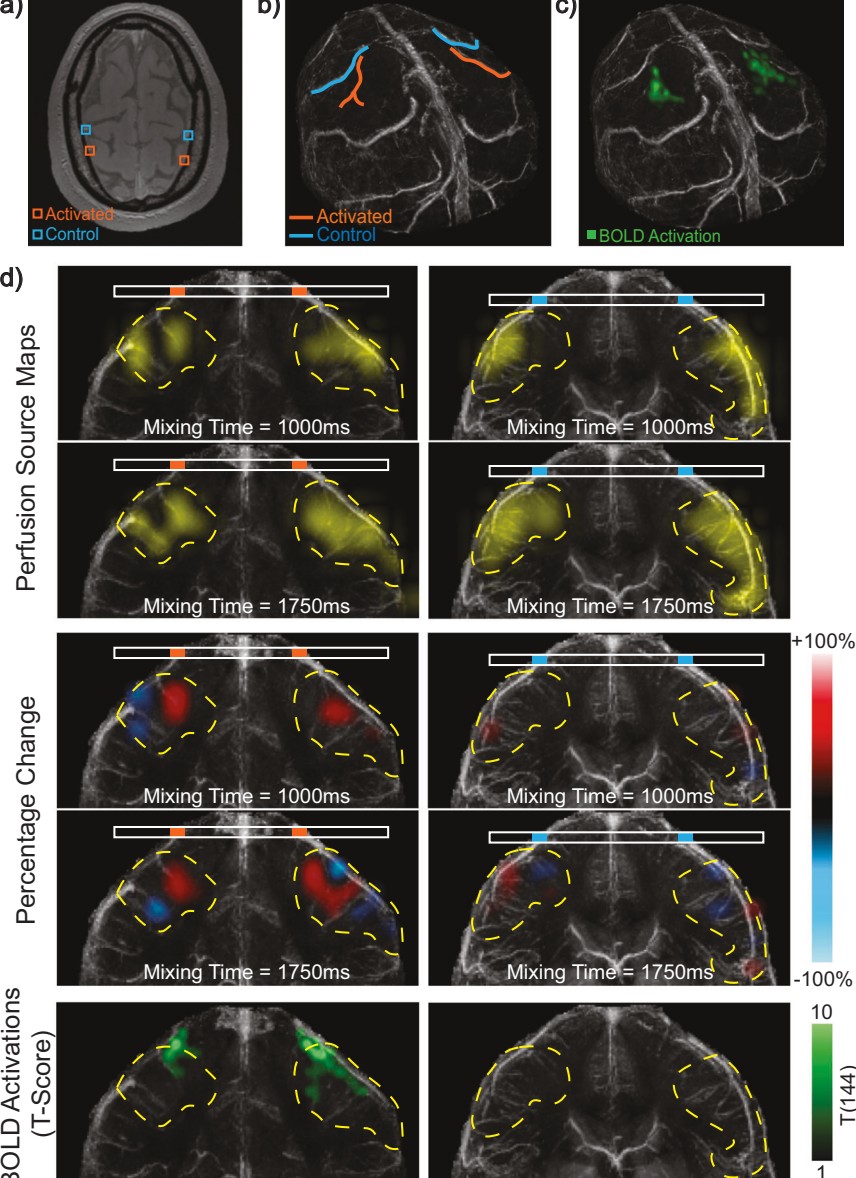

**Fig. 5 | Bilateral motor cortex activation in Subject 1. a** Imaging slice showing four selected veins: two (orange) drain the activated motor cortices and two (blue) do not and are used as control. **b** The color-coded veins overlaid on a 3D Quantitative Susceptibility Mapping (QSM) venogram. **c** Blood Oxygenation Level Dependent (BOLD) activations (green) overlaid on the 3D QSM-venogram showing bilateral motor cortex activation. **d** Top: Perfusion source maps for activated and control veins at two mixing times. Middle: The percentage change between task and baseline perfusion source maps showing mostly positive and some negative percentage change in veins draining the activated regions. Little or no percentage change is observed in the veins selected as control. Bottom: BOLD T-scores, from one-sided T-test, for the coronal sections containing each vein. Venous territories are marked with yellow dashed contours. Source data are provided in ref. 54.

## Multi-slice perfusion source mapping

In order to demonstrate a potential way to accelerate image acquisition, we leverage the concept of multiplexing to image at multiple locations sequentially. The experimental setup and results are illustrated in Fig. 9. Instead of imaging a single axial slice, three imaging slices were prescribed, each separated by 1.8 cm. When an image is acquired at each position, the tagged signal from the blood at that location is suppressed, preventing it from reaching the next imaging slice. This ensures that the sources at each imaging slice originate from a smaller FOV near the slice, reducing the scan time. Each imaging slice is acquired dynamically every 100 ms, faster than in previous experiments, to ensure that the tagged blood signal is eliminated before it reaches the next slice.

The target image resolution was $4 \times 4$ mm$^2$, and the in-plane perfusion source map resolution was $6 \times 6$ mm$^2$ (projected over AP). The perfusion source map FOV was reduced to $60 \times 36$ mm$^2$, resulting in an overall scan time of 12 min, three-fold faster than prior experiments.

For each superior vein, we used the QSM-venogram to identify the locations at each imaging slice that contain branches of that superior vein. The perfusion source maps from all branches of that vein were then combined and overlaid on the corresponding section of the QSM-venogram. The results for three different superior veins are shown in Fig. 9c. Since the images were acquired at a faster temporal resolution in this experiment, the perfusion source maps were averaged over two consecutive mixing times (400–500 ms, 800–900 ms, and 1200–1300 ms) and displayed at average mixing times of 450, 850, and 1250 ms.

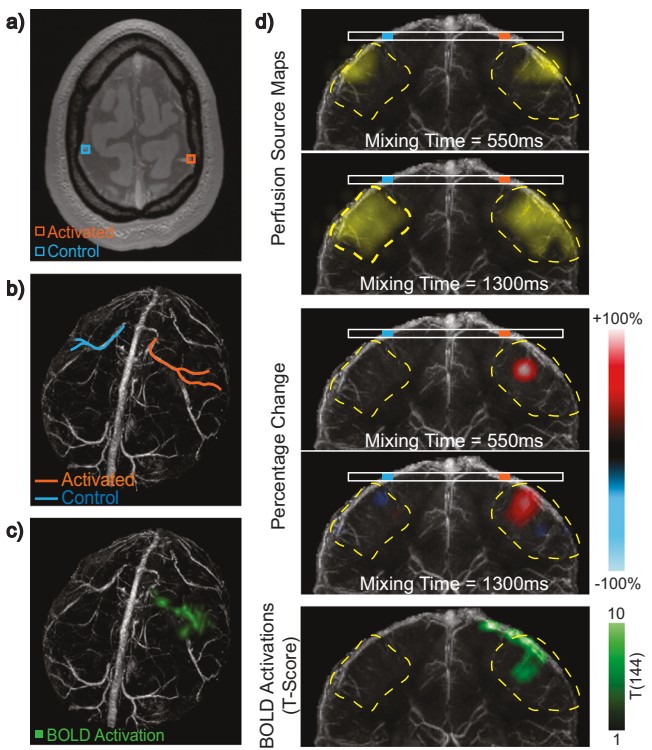

**Fig. 6 | Left motor cortex activation in Subject 2. a** Imaging slice showing two selected veins: one (orange) drains the activated left motor cortex and one (blue) is selected as control. **b** The color-coded veins overlaid on a 3D Quantitative Susceptibility Mapping (QSM) venogram. **c** Blood Oxygenation Level Dependent (BOLD) activations (green) overlaid on the 3D QSM-venogram. **d** Top: Perfusion source maps for activated and control veins at two mixing times. Middle: The percentage change between task and baseline perfusion source maps showing mostly positive local percentage change near the activated regions. Little or no percentage change is observed in the control vein. Bottom: BOLD T-scores, from one-sided T-test, for the coronal sections containing each vein. Venous territories are marked with yellow dashed contours. Source data are provided in ref. 54.

## Discussion

Perfusion source mapping with DiSpect is a highly versatile technique, which can probe the complex perfusion dynamics of the cerebral venous system. DiSpect's unique capability to trace the fluid pathway of blood spins across short to long time scales, regardless of the path and velocity, enables it to measure blood flow through a variety of vessels, ranging from capillary beds and smaller veins to larger sinuses. This approach is particularly well-suited to image veins where blood from smaller branches merge together to form larger flows, in contrast to arteries, where blood diverges and spreads into smaller branches. To investigate venous perfusion, blood water spins are labeled while in the smaller veins, capillaries and surrounding tissue and then detected once they drain into larger veins that cross a target plane near the brain's surface—where the signal-to-noise ratio is 3–4 times higher. By imaging at this location, where blood is quickly refreshed, we can dynamically trace the sources of blood draining into these veins from tissue outside the imaging slice without disrupting tissue perfusion.

Initially, we demonstrated venous perfusion source mapping by placing an axial slice towards the surface of the head, intersecting the superior sagittal sinus and several superior veins. We show that our method can dynamically resolve the sources of blood coming into the imaging slice. At shorter mixing times, the perfusion sources appear to be more focal, originating primarily from within large veins, whereas at longer mixing times, the perfusion sources appear more dispersed, as blood arrives from smaller veins and possibly from capillaries and surrounding tissue. For visualization, the perfusion source maps were

overlaid on QSM-venograms depicting the detailed structure of the veins. Notably, the perfusion source maps overlap very well with the underlying venous structure depicted by QSM.

To illustrate the capability to detect remote perfusion sources, we imaged a lower axial slice intersecting the straight sinus. Despite selecting an image ROI near the surface of the brain, we were able to resolve perfusion sources originating from deep structures near the center of the brain. These perfusion sources were detected from regions up to 15 cm away from the selected ROI.

In addition to its ability to trace venous perfusion dynamically, our method also exhibits sensitivity to global changes in venous perfusion due to caffeine. The perfusion source maps revealed a delayed arrival of blood to the imaging slice after caffeine intake. The delayed arrival was observed consistently for different veins as well as across different source locations within each venous territory. Interestingly, the delay in arrival time was longer towards the boundaries of the venous territories, further from the imaging slice, compared to regions closer to the imaging slice (Fig. 4e). This difference may be attributed to two factors. First, since it takes longer for blood to arrive from the boundaries, the absolute change in arrival time could be larger, even though the relative change is similar. Second, there could be effects due to the projection along the AP direction, which might be more pronounced near the edges where the veins branch out more.

Perfusion source mapping also shows specificity to local perfusion changes during a motor cortex activation task. In this case, we only observe local changes in the territories of veins that drain functionally activated regions. Even though imaging is performed at a location downstream of the activated area, our method can discern the blood within the imaged volume originating from an activated region upstream. The modulations within the territories are mainly located in the vicinity of BOLD activated regions, however, they are not identical to the BOLD activation. This is because the information provided by the perfusion source maps and BOLD are inherently different. The perfusion source maps offer dynamic information on the volume of blood flowing into a specific image voxel from a given source location at a particular time following tagging. In contrast, BOLD measures blood oxygenation changes that are also confounded by increases in blood flow and volume. BOLD contrast is affected by arteries, capillaries and veins, whereas the perfusion source maps are solely sensitive to capillaries and veins. Another difference is the resolution of the perfusion source maps (6–8 mm) compared to BOLD (3 mm). By varying the mixing times, perfusion source mapping can also provide information on the spatiotemporal modulation of venous perfusion due to brain activity.

In general, we observed an increase in perfusion source signal during task near regions showing functional activation. Interestingly, near the right motor cortex of Subject 1, we observed positive changes in source signal near the focal site of activation while observing significant negative changes in some veins slightly further away. This observation was consistent across two repeated scans, with the negative changes observed in the vicinity of less statistically significant BOLD activations. Previous literature has reported negative changes in venous blood volume during activation when using hyperoxia as a contrast agent in regions with positive changes in BOLD signal[38]. In addition, it has been reported that increased cerebral blood flow to a specific region during neural activation can lead to the redistribution of blood from surrounding vessels toward the activated region, a phenomenon known as arterial blood stealing[39]. We hypothesize that the observed negative changes in perfusion source signal may result from this compensatory mechanism. Further studies involving additional subjects are necessary to better understand and characterize this behavior.

For the repeatability analysis, the two scans were performed during different scan sessions, for Subjects 1 and 2, resulting in differences in head orientation. Consequently, due to this orientation

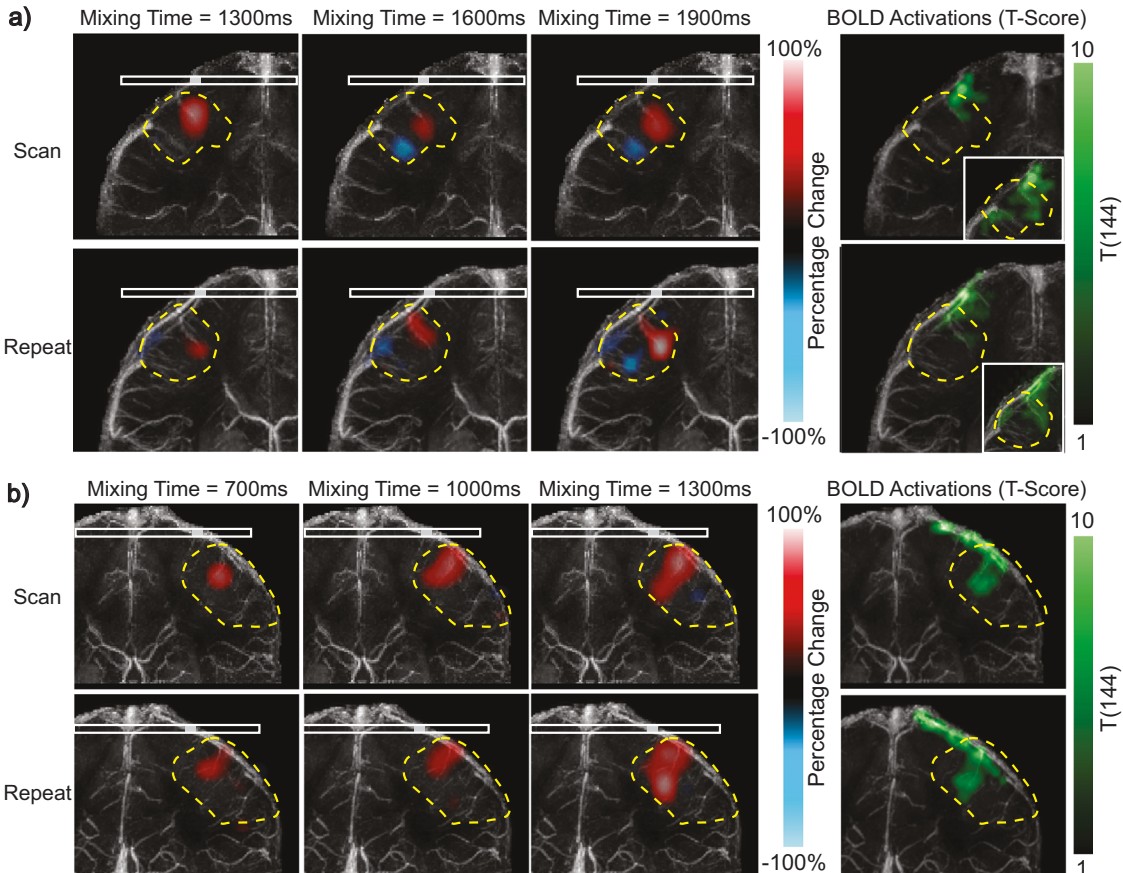

**Fig. 7 | Repeatability analysis for neural activation experiments of Subjects 1 and 2. a** Left: Percentage change in perfusion source maps at three mixing times for the vein draining the right motor cortex of Subject 1. Results are displayed for the initial and repeat scans. Right: Blood Oxygenation Level Dependent (BOLD) T-scores (green), from one-sided T-test, of the coronal section containing the vein. To visualize functionally activated regions in more detail, the BOLD T-scores are shown with a higher significance threshold on the right corner. **b** Left: Percentage change in perfusion source maps at three mixing times for the vein draining the left motor cortex of Subject 2. Right: BOLD T-scores (green), from one-sided T-test, of the coronal section containing the vein. The initial and repeat scans show good overlap for both subjects in regions that exhibit positive and negative percentage change. Note that the scans were performed on different sessions, leading to some differences in head orientation and imaging slice position. Venous territories are marked with yellow dashed contours. Source data are provided in ref. 54.

mismatch, the percentage change maps were not identical. Nevertheless, the regions of the venous territory with large perfusion modulation show significant overlap, supporting our claim that perfusion source mapping can repeatably detect modulations in venous perfusion due to neural activation.

To address the scan time limitations of our method, we employed the concept of multiplexing to accelerate the acquisition. In this experiment, we imaged three axial slices, effectively imaging perfusion sources at multiple locations sequentially. By combining the information from these slices, we were able to obtain the dynamics of entire superior veins. This approach reduced the scan time approximately by a factor of three, and experiments involving more slices could have the potential to further reduce scan time. Another main advantage of multi-slice imaging is that the blood comes from nearby sources, which reach the imaging slice at a shorter mixing time. This shorter mixing time reduces the amount of $T_1$ decay, preserving SNR. Despite these improvements, multi-slice perfusion source mapping has some limitations. Signals from smaller branches of veins with slow flow may be suppressed by the imaging slices, potentially missing some perfusion dynamics. Additionally, in our current implementation, the branches of each superior vein are identified manually using the QSM-venogram. Developing automated methods to determine the regions within each imaging slice that correspond to specific veins will enhance the reproducibility of the technique. Furthermore, increasing the number of slices would

require acquiring images at a faster temporal rate, which may introduce Specific Absorption Rate (SAR) limitations. Another limitation is that, unlike the original experiments where the imaging slice was close to the surface and benefited from high SNR, the additional slices in the multi-slice approach are farther from the surface and do not have this advantage.

One of the main advantages of DiSpect is its ability to image slower flows across long evolution times without any limitations due to gradient amplitude or field inhomogeneities. However, the SNR diminishes at longer evolution times due to the $T_1$ decay between tagging and imaging. Higher field strengths, i.e., 7T, could be advantageous as the longer $T_1$ of blood could allow the imaging of sources at longer mixing times after tagging. Moreover, physiological noise, especially due to the pulsatility of the static brain tissue, can play an important role at longer mixing times as additional cardiac cycles occur between tagging and imaging making the signal at longer mixing times inconsistent. Reducing physiological noise by periodically saturating signal from static tissue or detecting and accounting for the additional cardiac cycles is an area of future work.

In addition to the evolution time, there are several other parameters that impact the perfusion source map SNR. As in conventional imaging, the SNR is proportional to the size of the perfusion source map resolution. The SNR is also proportional to the square root of the number of displacement encodings, with more encodings effectively averaging the acquired signal, similar to the role of phase encodings in

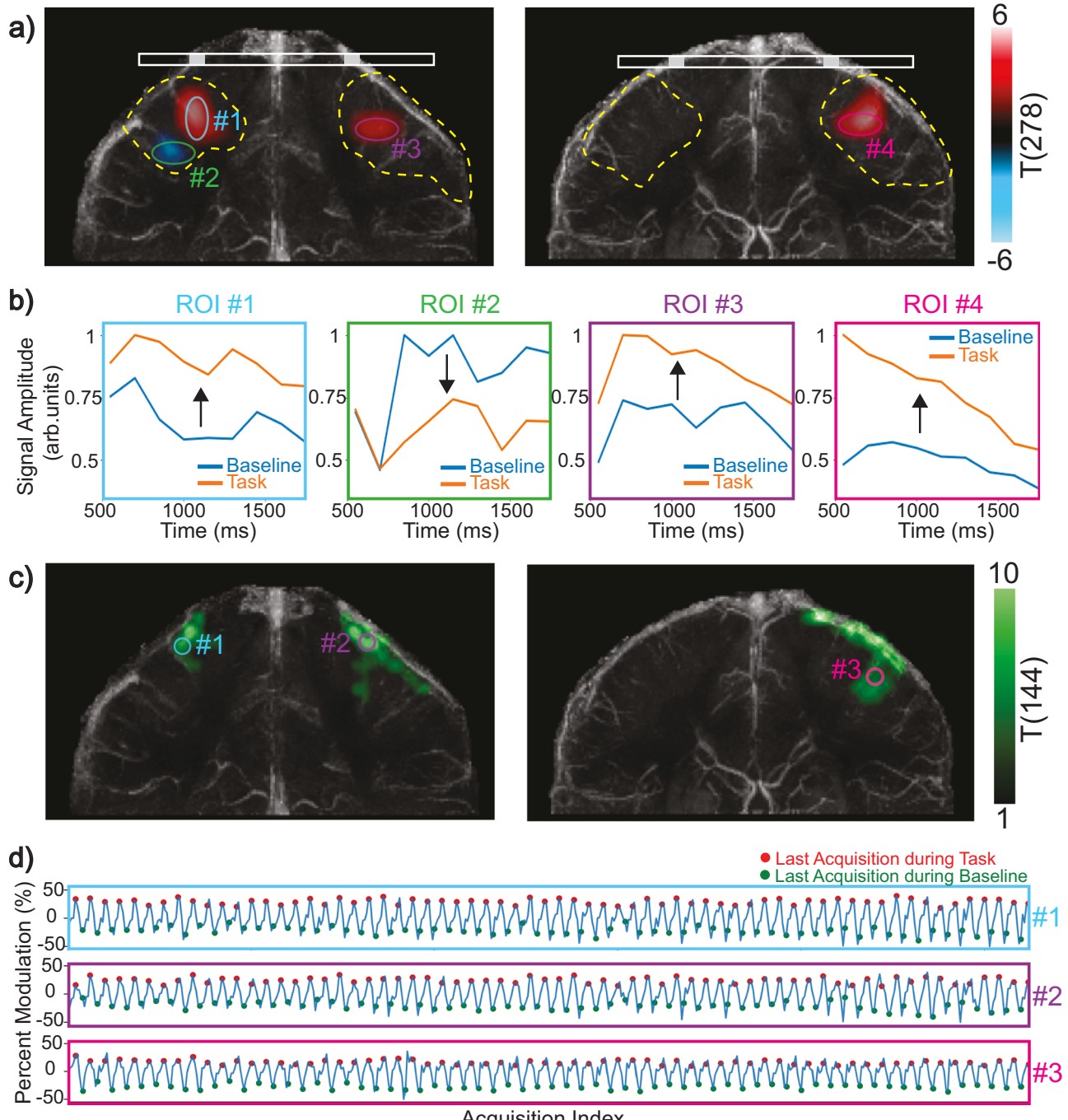

**Fig. 8 | Analysis of perfusion source map modulations due to neural activation.**
**a** Two-sided T-test results conducted on the perfusion source maps for Subject 1 with bilateral activation (left) and Subject 2 with left motor cortex activation (right). For Subject 1, regions with significant changes in source signal amplitude are identified in veins draining both motor cortices, whereas for Subject 2, significant changes are only identified in the vein draining the left motor cortex. **b** Four source ROIs are placed at regions showing statistically significant changes. The source signal amplitude of each ROI is displayed across mixing times at baseline and during task. **c** Blood oxygenation Level Dependent (BOLD) T-scores for the coronal sections containing each vein. Three ROIs are selected on the activated regions determined by the commercial BOLD acquisition. **d** The percent modulation for each ROI (shown in **c**) measured by the custom Spiral-BOLD performed between each DiSpect partition is displayed across the entire acquisition. Red markers indicate the last Spiral-BOLD acquisition performed during task, and green markers indicate the last Spiral-BOLD acquisition performed during rest. All three regions exhibit consistent activation throughout the scan. Source data are provided in ref. 54.

conventional imaging. Another key factor is the image voxel size; however, SNR depends not only on voxel size but on the percentage of the voxel volume occupied by the vein. This is due to the fact that the tagged signal solely comes from fresh blood entering the voxel and not from the static tissue in that voxel. In other words, as long as a single vein is entirely contained within the voxel, reducing the image voxel size does not reduce DiSpect SNR. Additionally, the rate at which images are acquired dynamically after tagging impacts SNR. If the timing between consecutive acquisitions is too short, the tagged blood signal in the imaging slice may not fully refresh, reducing SNR. These various parameters collectively impact the SNR and should be carefully considered.

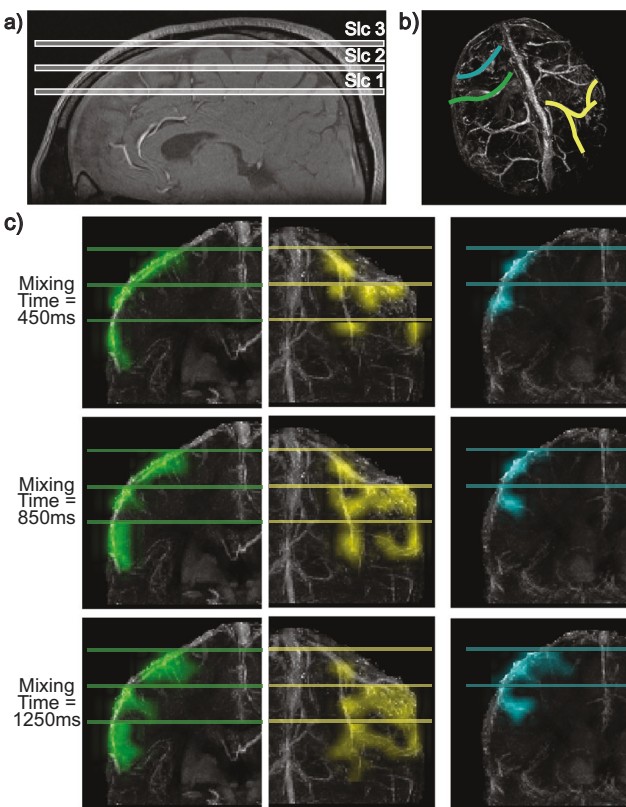

**Fig. 9 | Setup and results for multi-slice perfusion source mapping of superior veins. a** A sagittal slice showing the placement of the three prescribed axial imaging slices, separated by 1.8 cm. **b** A Quantitative Susceptibility Mapping (QSM) venogram highlighting the three selected superior veins in green, yellow, and blue. **c** For each selected superior vein, the locations in each slice containing branches of that vein were identified using the QSM-venogram and combined. The combined perfusion source maps were overlaid on the corresponding sections of the QSM-venogram, illustrating the dynamics of the selected veins. Perfusion source maps are displayed at average mixing times of 450 ms, 850 ms, and 1250 ms, derived by averaging across consecutive mixing times (400–500 ms, 800–900 ms, and 1200–1300 ms). The positions of the three imaging slices are also shown. While the two larger veins (green and yellow) have branches visible across all three imaging slices, the smaller vein (blue) is covered only by the top two slices. Source data are provided in ref. 54.

Another technique, VENTI[26], was the first to demonstrate the ability to map venous territories by spatially encoding tissue magnetization, similar to DiSpect, and using a phase-contrast angiographic readout to isolate intravascular signal. While VENTI can map venous territories, it does not provide any dynamic information on venous flow. In addition, the venous territories are resolved at a very low spatial resolution (2 cm). Furthermore, while VENTI resolves territories in the in-plane image dimensions, our approach resolves sources orthogonal to the imaging plane, enabling us to capture information across three dimensions. Finally, although the initial feasibility was demonstrated on a single subject, no further validation or analysis was performed.

Although perfusion source mapping with DiSpect is a highly versatile, rich and sensitive technique, it is currently constrained by several limitations. One limitation is that the acquisition of the high-dimensional dataset is time-consuming. For this reason, in this work, we have limited ourselves to acquiring 2D projections of perfusion source maps and low source resolutions (6–8 mm). These acquisitions required scan durations of 30–45 min, which is impractical for routine use. However, the dynamic perfusion source maps are highly sparse, and since flow is continuous, they exhibit substantial temporal and spatial redundancy. In addition, the sources of blood are constrained by the venous anatomy. We already demonstrated that acquisitions can be accelerated through sequence modification such as multi-slice acquisitions, reducing the scan time to 12 min. Future work could further accelerate acquisitions by using partial Fourier or variable density displacement k-space sampling, and a regularized reconstruction[40–42]. By combining acceleration through various different approaches, we aim to achieve significant reductions in scan time without significantly compromising SNR.

The visualization of our perfusion source maps could also be improved. Currently, we use a coronal projection of the QSM-venogram to visualize the venous anatomy of each cerebral vein. However, these projections may include contributions from neighboring veins. To enhance visualization, we are exploring advanced vessel segmentation methods to isolate the structure of each vein more effectively.

Despite these limitations, our innovative approach could open avenues for imaging various cerebral diseases and enhancing the success of existing imaging techniques. For example, it could facilitate the evaluation of arteriovenous malformations, where arteries and veins form abnormal, complex connections[43]. DiSpect also holds potential in investigating collateral circulation in ischemic stroke, particularly in cases involving complex vascular bypass networks within the venous drainage system[44]. Our tool could also assist in performing hemodynamic analysis of cerebral aneurysms to assess risk[45]. Additionally, our method could be adapted to explore the brain's glymphatic system, which is thought to play a critical role in clearing waste from the brain[46]. Since perfusion source maps can track blood sources regardless of velocity, they may be useful for measuring the very slow flows in the glymphatic system. We have already performed preliminary experiments to image flow in the ventricles and subarachnoid spaces, demonstrating the feasibility of our method to measure these slower flows[47]. Furthermore, our technique's ability to detect changes in venous perfusion during functional activation can contribute to a better understanding of venous contributions to the BOLD signal, thereby eliminating any mislocalization of functional activation and enhancing its spatial specificity.

## Methods
### Pulse sequence
The DiSpect pulse sequence begins with a DENSE tagging module, as illustrated in Fig. 2a. Imaging is performed repeatedly after tagging in order to sample tagged spins arriving at the imaging slice at increasingly longer mixing times. As in DENSE, each acquired image contains a displacement-encoded stimulated echo, with phase proportional to displacement occurring across the mixing time, along with two additional echoes that contribute to image artifacts[48]. To isolate the displacement-encoded stimulated echo, three-point phase cycling is applied to the second RF pulse of the tagging module[49]. DiSpect performs multiple acquisitions with incrementing DENSE encodings to capture a displacement spectrum. Essentially, DiSpect can be considered as spectral DENSE. The resolution of the displacement spectrum is determined by the maximum DENSE encoding, while the displacement field of view is defined by the step size of the DENSE encodings.

For this application, the source displacement spectra are called perfusion source maps, as they depict the sources of perfused blood from surrounding tissue, capillary beds and smaller veins that drain into a larger cerebral vein.

The perfusion source map of a target image voxel represents a map of spins that moved into that target voxel across a mixing time,

$t_{\text{mix}}$. The perfusion source maps are in relative coordinates to the location of the target voxel. The source map amplitude of a target voxel $\mathbf{r}$, displacing from a source position $\Delta\mathbf{r}$ away is:

$$s_{\mathbf{r}}(\Delta\mathbf{r}, t_{\text{mix}}) = M(\mathbf{r})p_{\mathbf{r}}(\Delta\mathbf{r}, t_{\text{mix}})e^{\frac{-t_{\text{mix}}}{T_1}}. \tag{1}$$

Here, $M(\mathbf{r})$ is a constant depending on the total amount of blood magnetization that moves into the target voxel, $p_{\mathbf{r}}(\Delta\mathbf{r}, t_{\text{mix}})$ is a function representing the ratio of blood that displaced into the voxel at time $t_{\text{mix}}$ from a source position $\Delta\mathbf{r}$ away, and $e^{\frac{-t_{\text{mix}}}{T_1}}$ is a term representing $T_1$ decay. Therefore, qualitatively, the signal intensity at a source location reflects the relative portion of blood that moved from that source location to the target image voxel. It is possible to convert the blood magnetization amplitude, $M(\mathbf{r})$, to the volume of blood using a separate calibration scan, which can be performed in the future to obtain quantitative values of the volume of blood.

## Image acquisition and parameters

All scanner experiments were conducted with a 3T GE MR750W system on healthy volunteers (GE Healthcare; Waukesha, Wisconsin). The DiSpect acquisition was implemented using RTHawk (Vista AI, Palo Alto, CA). Data collection was approved by the Institutional Review Board at UC Berkeley (CPHS #2010-07-1830). Written consent was obtained from all subjects. No monetary compensation was provided to participants. Experiments were performed on three male and two female subjects (average age: 39.8 years), with sex determined through self-reporting. Sex and age were not considered variables in this study because the research was focused on demonstrating the method.

**DiSpect.** For all experiments imaging the superior veins, 2D displacement encoding was performed along the LR and SI directions, resulting in coronal perfusion source maps. The axial imaging slice was placed close to the top of the head, where it intersects the superior sagittal sinus and several superior veins. For the experiment imaging the deep cerebral veins, 2D displacement encoding was performed along the AP and SI directions, resulting in sagittal perfusion source maps. The axial imaging slice was placed to intersect the straight sinus. For all DiSpect experiments, the target image resolution was $4 \times 4$ mm$^2$, FOV = $16 \times 16$ cm$^2$, flip angle = 90°, TE/TR = 1.8 ms/3 s and slice thickness = 5 mm. A 30-mm-thick axial spatial saturation band was placed around the imaging slice. Data were acquired at 19 mixing times between 100 ms and 3 s, in 150 ms increments.

For the initial experimental demonstration of DiSpect in the superior veins, a source displacement resolution of $6 \times 6$ mm$^2$ and a source FOV of $13.2 \times 8.4$ cm$^2$ were used. The overall scan duration was variable due to cardiac gating but was close to 45 min. This experiment was conducted on a male subject (49 years old). For the experiment imaging the deep cerebral veins, a source displacement resolution of $6 \times 6$ mm$^2$ and a source FOV of $12 \times 7.2$ cm$^2$ were used. The overall scan duration was variable due to cardiac gating but was close to 35 min. This experiment was conducted on a female subject (27 years old). To measure global perfusion changes induced by caffeine, a source resolution of $6 \times 6$ mm$^2$ and a source FOV of $12 \times 7.2$ cm$^2$ were used, resulting in a scan time of about 35 min. This acquisition was performed at baseline and 20 min after orally consuming 225mg of caffeine (GummiShot Energy Gummies, Santa Cruz, CA). This experiment was conducted on a male subject (49 years old).

In order to measure local perfusion modulation during neural activation, a total of four DiSpect acquisitions were performed on two male volunteers (average age: 47 years). For Subject 1, the first scan was performed with a source resolution of $8 \times 8$ mm$^2$ and a source FOV of $11.2 \times 8$ cm$^2$, while the subject was instructed to squeeze both hands. A second repeat scan was performed on a different day with a higher source resolution of $6 \times 6$ mm$^2$ and a source FOV of $7.2 \times 4.8$ cm$^2$, while

the subject was instructed to squeeze only their left hand. Subject 2 underwent the same protocol with a source resolution of $8 \times 8$ mm$^2$ and a source FOV of $11.2 \times 8$ cm$^2$ on two separate days while being instructed to squeeze their right hand. An additional female subject (49 years old), Subject 3, underwent a single scan with a source resolution of $8 \times 8$ mm$^2$ and a source FOV of $11.2 \times 8$ cm$^2$ while being instructed to squeeze their left hand. The total scan duration for the DiSpect acquisitions was approximately 50 min, which included 40 min for the partitioned DiSpect and an additional 10 min for spiral fMRI acquisitions performed between partitions.

**Multi-slice DiSpect.** For all multi-slice experiments, three axial imaging slices were prescribed, each separated by 1.8 cm. The slices were placed towards the top of the head to capture the superior veins. Imaging slices were acquired sequentially in a repeated order (slice 1, slice 2, slice 3, and back to slice 1), with a temporal gap of 33 ms between consecutive slice acquisitions. At the end of the acquisition, we obtain the perfusion source maps for each slice from 100 ms to 1.9 s in 100 ms intervals. The imaging parameters for multi-slice DiSpect included a target image resolution of $4 \times 4$ mm$^2$, an FOV of $21 \times 21$ cm$^2$, a slice thickness of 4 mm, a flip angle of 90°, and TE/TR = 1.8 ms/3 s. The perfusion source maps were obtained with a source displacement resolution of $6 \times 6$ mm$^2$ and a source FOV of $6 \times 3.6$ cm$^2$ for each slice. A 6-mm spatial saturation band was placed around imaging slices to suppress signals from static tissue. The total scan duration was approximately 12 min. This experiment was conducted on a male subject (29 years old).

**QSM.** For each scanned subject, QSM was used to obtain a detailed map of the veins. A Multi-Echo 3D-GRE sequence was performed with 16 echo times, spatial resolution = $0.8 \times 0.8 \times 0.8$ mm$^3$, TE1 = 2.12 ms, $\Delta$TE = 2.42 ms, TR = 41.0 ms, FOV = $21 \times 21 \times 13$ cm$^3$, and Flip Angle = 12°. We used STI Suite V3.0 (https://people.eecs.berkeley.edu/~chunlei.liu/software.html) to obtain susceptibility maps. Briefly, phase images at each echo time are unwrapped using Laplacian-based method[50]. The background field is filtered out by V-SHARP method[51] with SMV radius of 20 followed by iLSQR dipole inversion[52]. The obtained susceptibility maps from echoes 5–16 were averaged across echo times.

**BOLD fMRI.** For each scan session involving neural activation, a product 2D EPI-BOLD sequence was performed with 29 slices, resolution = $3.3 \times 3.3$ mm$^2$, FOV = $21 \times 21$ cm$^2$, TE = 28 ms and TR = 2 s. The subject was cued with vocal commands to perform a hand-squeezing task for 20 s on and 20 s off for a total duration of 5 min. For the custom Spiral-BOLD acquisitions conducted between DiSpect partitions, a lower image resolution of $4 \times 4$ mm$^2$ was used to reduce off-resonance artifacts. The remainder of the parameters were identical between the product EPI-BOLD and custom Spiral-BOLD acquisitions. The Spiral-BOLD acquisitions were performed for 5 TRs (10 s) at the beginning of each DiSpect partition.

## Data visualization and statistical analysis

**Spiral acquisition off-resonance correction.** Single-shot spiral acquisitions are susceptible to off-resonance artifacts due to their long readout durations. To correct for these artifacts, we employed a multi-frequency reconstruction approach. Images were reconstructed at frequency offsets ranging from −50 Hz to 50 Hz in 10 Hz intervals. For each location within the imaging slice, the frequency offset that yielded the sharpest image was qualitatively selected.

**Visualization of perfusion source maps.** For visualization, image voxels containing different cortical veins were selected from the imaging slice. The coronal perfusion source maps for each vein at different mixing times were overlaid on the corresponding venous anatomy. The coronal vein structure was visualized for each vein with a

maximum intensity projection of the echo-averaged susceptibility maps along the AP direction. The slab thickness for the projection was determined based on the extent of the vein along the AP direction. The perfusion source maps were interpolated to match the resolution of the susceptibility maps. Since the perfusion source map of each voxel is relative to the position of that voxel, the interpolated maps were shifted to match the coordinates of the susceptibility maps.

For multi-slice data, information from all three imaging slices was combined for visualization. For each superior vein, branches contained in different slices were identified manually using the QSM-venogram. The perfusion source maps from different branches were combined and overlaid on the venous anatomy of the corresponding sections in the QSM-venogram. To account for differences in slice positions, the maps from each slice were shifted appropriately to align with the global coordinate system of the venogram. The combined maps were visualized at average mixing times of 450 ms, 850 ms, and 1250 ms by averaging perfusion source maps across consecutive mixing times (e.g., 400–500 ms, 800–900 ms, 1200–1300 ms).

**Analysis of caffeine-induced global perfusion changes.** To visualize changes in perfusion source maps resulting from caffeine, we selected target image voxels corresponding to two cortical veins. We then picked two regions from the perfusion source maps of each cortical vein. The source signal amplitude in these regions at baseline and post-caffeine were normalized by the overall energy of the image voxel. Then, the normalized signal amplitude at each mixing time ($t_{mix}$) was corrected for $T_1$ decay, i.e., multiplied by $e^{\frac{t_{mix}}{T_1}}$ where $T_1 = 1584$ ms[53], and plotted. Next, to show the changes in venous arrival times, we first interpolated the mixing time signal series to a 15 ms time interval. For each source location, we identified the shortest mixing time at which the signal amplitude exceeded 50% of its maximum value across all mixing times. The value determined at baseline was subtracted from the post-caffeine value to determine the delay in arrival time.

**Analysis of local perfusion changes due to neural activation.** In order to visualize regions of the venous territories that show local perfusion modulation, we calculate the percentage signal change in perfusion source maps between baseline and task. The percentage signal change is calculated as $\frac{(s_{task} - s_{baseline})}{s_{baseline}} \times 100\%$, where $s_{baseline}$ and $s_{task}$ correspond to the perfusion source map signal during baseline and task, respectively. Two-sided T-tests were conducted with a significance threshold of $p = 0.05$ to compare the source mixing time signal series at baseline and during task for every point of the perfusion source maps. Regions of the perfusion source map with significant differences were identified, and the T-statistics were displayed.

The EPI BOLD datasets were preprocessed using SPM12 (https://www.fil.ion.ucl.ac.uk/spm/) by performing realigning, slice-timing correction, co-registration (to QSM acquisition) and smoothing (kernel size = $3 \times 3 \times 3$ mm³). One-sided T-tests were conducted to obtain T-statistic images with a cluster significance threshold of $p = 0.01$ and cluster extent threshold of 10. To visualize the functional activation in the right motor cortex of Subject 1 in more detail, T-statistic images were obtained with a higher cluster significance threshold of $p = 0.05$ and cluster extent threshold of 3.

### Reporting summary
Further information on research design is available in the Nature Portfolio Reporting Summary linked to this article.

## Data availability
All main data used and analyzed in this study are openly available in the Zenodo repository at https://doi.org/10.5281/zenodo.15178447[54].

## Code availability
The code needed to reproduce the figures in this manuscript is openly available at https://github.com/mikgroup/Venous-Perfusion-Source-Mapping[55].

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

## Acknowledgements

We would like to acknowledge Prof. Alex Pines, who recently passed away, whose research on remote NMR detection inspired this work. This work was supported in part by the NIH under grants R01MH127104 (E.K. and J.M.) and R01EB034831 (E.K.). Additional support was provided by the UCSF-UCB Collaboration Program (E.K.).

## Author contributions

E.K., Z.Z. and M.L. contributed to method design and development. E.K. conducted all scanner experiments. C.L. and M.L. advised on the design of experiments. J.C. developed pipeline for obtaining QSM-venograms. J.M. designed hardware used in experiments. All authors contributed to reviewing and editing the manuscript.

## Competing interests

The authors declare no competing interests.
