## [Transparent Peer Review file · Nature Communications]

MR Perfusion Source Mapping Depicts Venous Territories and Reveals Perfusion Modulation during Neural Activation

Corresponding Author: Dr Ekin Karasan

Version 0:

Reviewer comments:

Reviewer #1

(Remarks to the Author)

Thank you for the opportunity to review the article “MR Perfusion Source Mapping “In Reverse”: Maps Venous Territories and Reveals Perfusion Modulation during Neural Activation” by Karasan et al.

The article presents a relatively new concept for imaging the displacement of venous blood using MRI. The authors show that their method is sensitive to changes in venous blood volume resulting from brain activity as part of the BOLD response. The potential impact of the article is relatively high although the proposed applications are somewhat vague, but that is not a major concern. In its current implementation the method yields low resolution images, but the method is still fairly novel and there is significant potential for acceleration and increase in resolution.

In general, I am very positive about the article but have some minor comments that would like to see addressed.

While this is a very interesting new method, I think that there are some limitations that should be mentioned in the article. The displacement encoding is a net displacement during the encoding period, not the mixing time, and doesn't say much about the rest of the trajectory – I guess you have to infer that from the venogram.

Related, the encoding is also a first order moment – ie – no acceleration effects are considered and it may constitute a source of error.

Page 12; line 523: “ We hypothesize that the negative changes in perfusion source signal could be a result of the redistribution of venous blood volume from neighboring regions to the activated area.” I think the statement is not helpful as written. Perhaps better to mention the references to shunting ‘blood stealing’ in the arterial side, which would result in reduced venous blood in the region (e.g.- doi.org/10.1016/j.jtbi.2021.110856).

Some editorial comments:

Regarding the pulse sequence depicted in figure 2a – is the spatial saturation in the green box applied over the whole volume or only at the readout slice?

Page 13: line 527: the use/need for the ‘three-point phase cycling scheme’ is not obvious to this reader – please explain why this is necessary.

Reviewer #2

(Remarks to the Author)

Summary: The present study utilizes displacement spectrum MRI to perform venous source mapping, whereby imaging is performed near the brain's surface where SNR is high, and spectral displacement encoding using stimulated echoes is used to determine the spatial origins of the water from selected vein regions. While the imaging plane may be axial, the source mapping plane would typically be a coronal plane. Venous source mapping may be performed using a range of mixing times, ranging from 10ms to 3s, and is limited on the upper end by the T1 of venous blood. To display the low-resolution

source map images, they are overlaid on 2D projections of 3D venograph images which show the 3D anatomy of the large veins. The venous source signals are shown to be high in the vicinity of the large veins, with signal originating from within the large veins and from other areas in the vicinity of the large veins. A number of experiments are performed to demonstrate the technique. First, it is demonstrated that the venous source regions are larger as the mixing time increases. Next, it is shown that venous draining kinetics after caffeine are altered, as delayed arrival of water protons to surface veins are measured. Finally, detection of bilateral and left motor cortex activation are demonstrated, with excellent agreement between spectral displacement imaging and conventional BOLD MRI. The demonstrations of the capabilities of the novel method are compelling and suggest that the method may be applied to a variety of brain disorders in the future to potentially make new observations and discoveries. However, scan times apparently on the order of 40 minutes are a limitation that may hinder wider use.

Major comments:

1. The authors should discuss quantification. For example, in Figure 3, how can the intensity of the yellow venous source signal be interpreted in physical units or percentages?
2. The scan time for the neural activation studies is unclear. Please clarify. Did the subjects perform hand squeezing, in a partitioned manner as described, for 30-40 minutes?
3. The stated scan times of 35-45 minutes are quite lengthy, and at the same time the source maps have low spatial resolution in two dimensions and don't resolve the third dimension. To what extent will these limitations ultimately render the method to be quite impractical, or will acceleration and denoising techniques be able to overcome them? Is it anticipated that opportunities to accelerate exist in many dimensions such as the image spatial dimensions, the displacement-encoding dimensions, and the mixing time dimension? Is SNR high enough to support acceleration rates on the order of 10-fold or more?
4. Related to question 3, the authors should discuss generally which factors determine the signal-to-noise ratio of the venous source images, given that there are many factors including the image voxel size as well as the pixel size of the source maps.
5. I would speculate that until substantially accelerated versions of the method are developed, due to long scan times on the order of 40 minutes, utilization will be quite limited. Do the authors disagree?

Minor comments:

1. In the Methods section, citing prior work on DENSE is appropriate; however, I don't see a reason to refer to SPAMM tagging. The present method is utilizing properties of the stimulated echo, and the method can be exactly described using language about stimulated echoes and other echoes. Invoking SPAMM tagging seems tangential to the main idea. Also, DiSpect is a bit unusual and may not be an ideal name for the method. I wonder if spectral displacement encoding with stimulated echoes (spectral DENSE) might be a better name?
2. Is ECG gating necessary? Is venous flow in the brain strongly or just slightly pulsatile?
3. Do the authors think there is potential to investigate the brain's glymphatic system with this method? Perhaps they can speculate briefly on this point in the Discussion.
4. In the section on "Image acquisition and parameters", the number of mixing times is not provided.
5. In Fig. 2, do the successive "single shot readouts" represent different mixing times? If so, then the annotation of mixing time should reflect this.
6. Because of water exchange between the extra- and intra-vascular spaces, instead of discussing venous water as originating in capillaries, perhaps originating in capillaries and tissue might be more accurate. Perhaps also from the arterial system for longer mixing times.

Reviewer #3

(Remarks to the Author)

In the manuscript "MR Perfusion Source Mapping "In Reverse": Maps Venous Territories and Reveals Perfusion Modulation during Neural Activation" the authors Karasan et al apply their previously published technique 'DiSpect' to measure venous territories. They demonstrate the technique in healthy volunteers. Moreover, they measure changes in DiSpect signal during a caffeine challenge (N=1) and during a motor task (N=2). Whereas the DiSpect technique was published before in Magnetic Resonance in Medicine, the innovative nature of the current contribution is to measure the venous territories (the MRM paper was focused on the arterial side, where there are many more techniques to obtain similar information in much shorter scan-times). As the DiSpect technique is based on 2D encoding of the whole brain as well as encoding of the readout image, the technique is very slow. This results in long scan-times (30-40 minutes), which is seriously limiting the application opportunities of this approach. Whereas the technique does provide unique information, this is at the moment limited to information that is not very relevant to improve our understanding of the brain. On the positive side, this is a very innovative approach, a technical tour-de-force, and the experiments are presented in a very convincing manner. Beside these general comments, I have several concerns:

Main concerns:

1. In the analysis of the motor activation, the authors seem to double-dip: first they determine T-maps where there is a change in DiSpect signal, and subsequently they select ROIs based on these T-maps to look into the DiSpect signal. Not surprisingly they find changes in DiSpect signal. This should be analyzed in a different manner.
2. The authors claim an important role of veins in blood flow regulations. In my opinion the literature is more supportive of a passive role of the veins (following e.g. the thoracic pressure) with active control in arterioles and to a lesser extent by pericytes on capillaries. Please add references to support their claim.

3. The authors do not discuss the ISMRM-abstract by Eric Wong who was the first to measure venous territories. I am aware this has never been published as full paper, but credits should be given to him and a discussion on the differences with his technique would be beneficial.
4. When introducing DiSpect the authors should already mention that the signal decays with the T1 of blood.
5. Remote detection (page 3): the authors almost exclusively show cortical origins of the detected signal, i.e. also in the high SNR part of the head coil. Please reconsider this argument.
6. Figure 1: From the figure it is not clear that "n" refers to a 2D spatial-encoding scheme. This could be made more clear. Similarly the repeated images, do not give an impression that these are spatially encoded maps to discern the origin of their signal. Would be great when they make this figure easier to understand.
7. Figure 2A: it is not clear to what area the spatial saturation is applied. Also, include the SPAMM module explicitly in the diagram.
8. Page 13 (Discussion): please be open, how (unpractical) long the sequences did take. Please quote the scan-duration in this paragraph (near line 568).
9. Methods: this has quite some overlap with the MRM paper. Consider shortening.

Minor concerns:

10. Page 2, line 35: the venous signal change of BOLD is the main contributor to the mismatch between neuronal activation and BOLD response (so not "in addition...").
11. ASL and especially velocity-selective ASL has been used to select venous signal, also in the context of angiography (although then venous signal is often seen as something that should be avoided, making people to apply acceleration selective ASL instead).
12. Page 7, line 363: "...increase perfusion source map signal". A bit more help how to interpret this would be good.

Version 1:

Reviewer comments:

Reviewer #1

(Remarks to the Author)

the authors have addressed my concerns adequately. Thank you

Reviewer #2

(Remarks to the Author)

The author's responses fully addressed all of my previous comments, and I have no further critiques.

Reviewer #3

(Remarks to the Author)

I thank the authors for their revisions of their manuscript. I will only focus on remaining concerns:

- 1) Following up on my second main concern: please consider rephrasing the first sentence of the manuscript. This still seems to imply a much more active role of the venous system (although I of course recognize that the venous system is critical).
- 2) Follow-up on point 8: Although the multislice experiment in 12 minutes is a step in the right direction, it is still very far from whole brain scanning in a feasible scantime. Therefore, this manuscript remains in the very interesting technical approach with very little clinical impact. Or can the authors include a clinical example showing the added value of their approach?

Response to Reviewers

MR Perfusion Source Mapping In Reverse: Maps Venous Territories and Reveals Perfusion Modulation during Neural Activation

Ekin Karasan Jingjia Chen Julian Maravilla Zhiyong Zhang Chunlei Liu
Michael Lustig

We would like to thank our editor and the three referees for their thorough review and suggestions, which have significantly improved the quality of our paper. We revised our paper accordingly in order to alleviate all of the concerns. Below we list the reviewer's comments followed by our detailed responses in blue.

Comments of Reviewer 1 :

Thank you for the opportunity to review the article "MR Perfusion Source Mapping "In Reverse": Maps Venous Territories and Reveals Perfusion Modulation during Neural Activation" by Karasan et al.

The article presents a relatively new concept for imaging the displacement of venous blood using MRI. The authors show that their method is sensitive to changes in venous blood volume resulting from brain activity as part of the BOLD response. The potential impact of the article is relatively high although the proposed applications are somewhat vague, but that is not a major concern. In its current implementation the method yields low resolution images, but the method is still fairly novel and there is significant potential for acceleration and increase in resolution.

In general, I am very positive about the article but have some minor comments that would like to see addressed.

We sincerely thank you for your thoughtful and constructive feedback, as well as your positive remarks. We greatly appreciate the time and effort.

While this is a very interesting new method, I think that there are some limitations that should be mentioned in the article. The displacement encoding is a net displacement during the encoding period, not the mixing time, and doesn't say much about the rest of the trajectory – I guess you have to infer that from the venogram.

We apologize for the confusion. As shown in the pulse sequence in Figure 2a, during tagging, the information about the source positions of spins is instantaneously encoded into the longitudinal magnetization during the short encoding period. Following this, spins displace across the mixing time (t_{mix}), and the displacement information is decoded once the spins reach the imaging slice. Therefore, the displacement period is actually the mixing time, similar to DENSE [28, 29]. Since we perform imaging repeatedly after tagging, we effectively measure displacements accumulated across different mixing times ($t_{\text{mix}_1}, t_{\text{mix}_2}, t_{\text{mix}_3}, \dots$). We clarified this point in the Methods section (Page 16, Lines 857-877). Further details of the DiSpect method are provided in [27].

Related, the encoding is also a first order moment – ie – no acceleration effects are considered and it may constitute a source of error.

DENSE/DiSpect encodes the position and not the velocity of spins in M_z . It is true that we did not account for velocity or acceleration effects (first- and second-order moments). However, the gradients used for displacement encoding in our method are very low, with the highest gradient area being 0.83 cycles/cm. The maximum velocity expected in the veins within the imaging slice is below 10 cm/s. For the gradients we use, a velocity of 10 cm/s would result in a phase shift of less than 1 degree, corresponding to a displacement error of approximately 0.03 mm. Given this minimal error, the effects of higher-order moments are negligible.

Page 12; line 523: "We hypothesize that the negative changes in perfusion source signal could be a result of the redistribution of venous blood volume from neighboring regions to the activated area." I think the statement is not helpful as written. Perhaps better to mention the references to shunting 'blood stealing' in the arterial side, which would result in reduced venous blood in the region (e.g.-doi.org/10.1016/j.jtbi.2021.110856).

Thank you for your valuable suggestion. To address your comment, we revised the corresponding section of the manuscript to incorporate this concept and cite the suggested reference. The updated text now reads (Page 14, Lines 650-658):

“In addition, it has been reported that increased cerebral blood flow to a specific region during neural activation can lead to the redistribution of blood from surrounding vessels toward the activated region, a phenomenon known as arterial blood stealing [39]. We hypothesize that the observed negative changes in perfusion source signal may result from this compensatory mechanism. Further studies involving additional subjects are necessary to better understand and characterize this behavior.”

Some editorial comments:

Regarding the pulse sequence depicted in figure 2a – is the spatial saturation in the green box applied over the whole volume or only at the readout slice?

To clarify, in our experiments, the spatial saturation band is positioned around the imaging slice to suppress signals from static tissue. We updated the caption of Figure 2 to explicitly state this.

Page 13: line 527: the use/need for the ‘three-point phase cycling scheme’ is not obvious to this reader – please explain why this is necessary.

In DENSE/DiSpect the acquired signal is a combination of three distinct echoes:

1. The displacement-encoded stimulated echo, which has a phase proportional to the displacement.
2. The complex conjugate of the displacement-encoded echo.
3. A T1-relaxation echo.

To extract the displacement-encoded stimulated echo and eliminate artifacts from the other two echoes, we perform phase cycling. In our original DiSpect paper, we employed a 4-point phase cycling scheme, as described in [48]. In this work, we use a 3-point phase cycling scheme, as outlined in [49], to reduce scan time. The reference for 3-point phase cycling was added to the paper for clarification (Page 16, Line 844).

Comments of Reviewer 2:

Summary: The present study utilizes displacement spectrum MRI to perform venous source mapping, whereby imaging is performed near the brain's surface where SNR is high, and spectral displacement encoding using stimulated echoes is used to determine the spatial origins of the water from selected vein regions. While the imaging plane may be axial, the source mapping plane would typically be a coronal plane. Venous source mapping may be performed using a range of mixing times, ranging from 10ms to 3s, and is limited on the upper end by the T1 of venous blood. To display the low-resolution source map images, they are overlaid on 2D projections of 3D venograph images which show the 3D anatomy of the large veins. The venous source signals are shown to be high in the vicinity of the large veins, with signal originating from within the large veins and from other areas in the vicinity of the large veins. A number of experiments are performed to demonstrate the technique. First, it is demonstrated that the venous source regions are larger as the mixing time increases. Next, it is shown that venous draining kinetics after caffeine are altered, as delayed arrival of water protons to surface veins are measured. Finally, detection of bilateral and left motor cortex activation are demonstrated, with excellent agreement between spectral displacement imaging and conventional BOLD MRI. The demonstrations of the capabilities of the novel method are compelling and suggest that the method may be applied to a variety of brain disorders in the future to potentially make new observations and discoveries. However, scan times apparently on the order of 40 minutes are a limitation that may hinder wider use.

We sincerely thank you for your time and effort. We greatly appreciate the kind words and constructive feedback.

Major comments:

1. The authors should discuss quantification. For example, in Figure 3, how can the intensity of the yellow venous source signal be interpreted in physical units or percentages?

To address this, we added a discussion of the signal quantification in the context of the results presented in Figure 3. Specifically, we explain how the signal intensity in the perfusion source maps relates to the relative contribution of blood from source regions that drains into the target imaging voxel. The added paragraph in the Results section reads (Page 5, Lines 268-282)

“The signal intensity in the perfusion source maps reflects the relative proportion of blood from each source position that has displaced into the target imaging voxel over a mixing time period. For example, in Figure 3b, the yellow perfusion source maps represent the proportion of blood that has displaced into the target voxel shown with a yellow box. While outside of the scope of this work, we expect that in order to obtain absolute quantitative maps, a calibration scan will be needed. This calibration scan will estimate the absolute volume of blood displacing into the image voxel, from which quantitative source maps could be computed. (see Methods section for more details on quantification).”

Additionally, the Methods section includes a more detailed explanation of how the perfusion source map intensity could be quantitatively interpreted in future work (Page 16, Lines 857-877).

2. The scan time for the neural activation studies is unclear. Please clarify. Did the subjects perform hand squeezing, in a partitioned manner as described, for 30-40 minutes?

The DiSpect acquisition for the neural activation studies was approximately 50 minutes, which included 40 minutes for the partitioned DiSpect and an additional 10 minutes for spiral fMRI acquisitions performed between partitions. We added this information in Page 17, Lines 939-944.

3. The stated scan times of 35-45 minutes are quite lengthy, and at the same time the source maps have low spatial resolution in two dimensions and don't resolve the third dimension. To what extent will these limitations ultimately render the method to be quite impractical, or will acceleration and denoising techniques be able to overcome them? Is it anticipated that opportunities to accelerate exist in many dimensions such as the image spatial dimensions, the displacement-encoding dimensions, and the mixing time dimension? Is SNR high enough to support acceleration rates on the order of 10-fold or more?

5. I would speculate that until substantially accelerated versions of the method are developed, due to long scan times on the order of 40 minutes, utilization will be quite limited. Do the authors disagree?

We thank you for these important comments, and respond to items (3) and (5) together, since they are related. We agree that acceleration is essential to make this method practical and widely usable. Without acceleration, the lengthy scan times would limit its clinical applicability.

The primary bottleneck in terms of sequence duration is the displacement encoding steps, and we believe that most acceleration opportunities lie in this dimension. The perfusion source maps are inherently sparse, and due to the continuous nature of blood flow, they also exhibit significant spatial and temporal redundancy across the mixing time dimension. Moreover, the sources of blood are constrained by the venous anatomy. Future work could leverage these properties to accelerate acquisitions by employing partial Fourier or variable density sampling techniques, along with regularized reconstruction methods. Furthermore, reducing the number of phase cycles required for echo separation, by utilizing prior knowledge about the T_1 echo and the conjugate echo, represents another avenue for acceleration.

We acknowledge that achieving large acceleration factors (e.g., 10x) with these methods alone might not be feasible, as it could detrimentally affect the SNR. To address this, we introduced a multi-slice imaging approach using the concept of multiplexing as a step toward accelerating DiSpect. For each DiSpect encoding instead of imaging a single axial slice, we acquired three. Each slice separated by 1.8 cm. When an image is acquired at each position, the tagged signal from the blood at that location is suppressed, preventing it from reaching the next imaging slice. This effectively limits the field of view for perfusion sources to a localized region around each slice. Data from all slices were then combined to reconstruct the full perfusion dynamics of the entire superior veins. This implementation reduced the total scan time by one-third (approximately 12 minutes) while maintaining the ability to resolve venous perfusion dynamics. One advantage of multi-slice imaging is that the blood comes from nearby sources, which reach the imaging slice at a shorter mixing time. This shorter mixing time reduces the amount of T1 decay, preserving SNR. The multi-slice experiment and results are described in Pages 10-11, Lines 501-536.

In summary, we anticipate that we can combine various acceleration strategies to achieve significant reductions in scan time while preserving SNR. For example, it is not unreasonable that in a future work we could achieve $\sim 2x$ acceleration using partial Fourier, another $\sim 2x$ from variable density sampling and spatio-temporal regularization, and another 1.5x by reducing the number of phase cycles. With the addition of the multi-slice approach we could eventually achieve an order of magnitude reduction in acquisition time. Based on the answer to item (4), a 6-fold acceleration of encodes would reduce source SNR by $\sim \sqrt{6} = 2.5$, which could be partially offset by partial-fourier positivity constraint and spatio-temporal regularization. The multi-slice approach can also boost SNR by reducing T1 decay. Therefore we anticipate sufficient SNR even at 10x accelerations.

4. Related to question 3, the authors should discuss generally which factors determine the signal-to-noise ratio of the venous source images, given that there are many factors including the image voxel size as well as the pixel size of the source maps.

We added a detailed discussion of the factors influencing the SNR of the venous perfusion source maps on Pages 14-15, Lines 725-749:

“In addition to the evolution time, there are several other parameters that impact the perfusion source map SNR. As in conventional imaging, the SNR is proportional to the size of the perfusion source map resolution. The SNR is also proportional to the square root of the number of displacement encodings, with more encodings effectively averaging the acquired signal, similar to the role of phase encodings in conventional imaging. Another key factor is the image voxel size; however, SNR depends not only on voxel size but on the percentage of the voxel volume occupied by the vein. This is due to the fact that the tagged signal solely comes from fresh blood entering the voxel and not from the static tissue in that voxel. In other words, as long as the image voxel is larger than the blood vessel it contains, reducing the image voxel size does not reduce DiSpect SNR. Additionally, the rate at which images are acquired dynamically after tagging impacts SNR. If the timing between consecutive acquisitions is too short, the tagged blood signal in the imaging slice may not fully refresh, reducing SNR. These various parameters collectively impact the SNR and should be carefully considered.”

Minor comments:

1. In the Methods section, citing prior work on DENSE is appropriate; however, I don't see a reason to refer to SPAMM tagging. The present method is utilizing properties of the stimulated echo, and the method can be exactly described using language about stimulated echoes and other echoes. Invoking SPAMM tagging seems tangential to the main idea. Also, DiSpect is a bit unusual and may not be an ideal name for the method. I wonder if spectral displacement encoding with stimulated echoes (spectral DENSE) might be a better name?

Based on your recommendation, we use terminology of stimulated echoes, rather than SPAMM tagging in the manuscript. We agree that “spectral DENSE” would have been a suitable alternative to DiSpect. However, since we rely on our previous work which already named it DiSpect [27], we prefer to keep the name. When mentioning DiSpect in the paper we added that it can also be thought of as “spectral DENSE” (Page 16, Lines 847-848).

2. Is ECG gating necessary? Is venous flow in the brain strongly or just slightly pulsatile?

This is a good point! The short answer is: it depends on the level of artifacts and physiological noise that would be acceptable. There are various studies (e.g. <https://doi.org/10.3389/fnins.2020.00415> and <https://doi.org/10.1002/jmri.1880040108>) that report pulsatility in the superior sagittal sinus and smaller cortical veins. To ensure that venous flow is consistent across our acquisitions, we use cardiac triggering. In addition, the brain tissue and CSF are also pulsatile, which sometimes lead to inconsistencies during echo separation and artifacts if cardiac triggering is not used – lowering the overall SNR. We chose triggering to get the most consistent data. Future studies could evaluate the tradeoffs of alleviating the need for triggering

3. Do the authors think there is potential to investigate the brain's glymphatic system with this method? Perhaps they can speculate briefly on this point in the Discussion.

We are definitely considering the glymphatic system as an application of DiSpect. We added the following section to the Discussion section (Page 15, Lines 813-823):

“Additionally, our method could be adapted to explore the brain’s glymphatic system, which is thought to play a critical role in clearing waste from the brain [46]. Since perfusion source maps can track blood sources regardless of velocity, they may be useful for measuring the very slow flows in the glymphatic system. We have already performed preliminary experiments to image flow in the ventricles and subarachnoid spaces, demonstrating the feasibility of our method to measure these slower flows. [47]”

4. In the section on “Image acquisition and parameters”, the number of mixing times is not provided.

Thank you for realizing this. We added the following sentence to that section (Page 16, Lines 903-904):

“Data were acquired at 19 mixing times between 100ms and 3s, in 150ms increments.”

5. In Fig. 2, do the successive “single shot readouts” represent different mixing times? If so, then the annotation of mixing time should reflect this.

The successive single-shot readouts represent different mixing times. To address your suggestion, we revised Figure 2a to mark the first two mixing times as $t_{\text{mix}1}$ and $t_{\text{mix}2}$. We also modified the corresponding figure caption to make this point more clear.

6. Because of water exchange between the extra- and intra-vascular spaces, instead of discussing venous water as originating in capillaries, perhaps originating in capillaries and tissue might be more accurate. Perhaps also from the arterial system for longer mixing times.

Thank you for the suggestion. We revised multiple sections throughout the manuscript to indicate that the blood water could originate from capillary beds as well as surrounding tissue.

Comments of Reviewer 3:

In the manuscript “MR Perfusion Source Mapping “In Reverse”: Maps Venous Territories and Reveals Perfusion Modulation during Neural Activation” the authors Karasan et al apply their previously published technique ‘DiSpect’ to measure venous territories. They demonstrate the technique in healthy volunteers. Moreover, they measure changes in DiSpect signal during a caffeine challenge (N=1) and during a motor task (N=2). Whereas the DiSpect technique was published before in Magnetic Resonance in Medicine, the innovative nature of the current contribution is to measure the venous territories (the MRM paper was focused on the arterial side, where there are many more techniques to obtain similar information in much shorter scan-times). As the DiSpect technique is based on 2D encoding of the whole brain as well as encoding of the readout image, the technique is very slow. This results in long scan-times (30-40 minutes), which is seriously limiting the application opportunities of this approach. Whereas the technique does provide unique information, this is at the moment limited to information that is not very relevant to improve our understanding of the brain. On the positive side, this is a very innovative approach, a technical tour-de-force, and the experiments are presented in a very convincing manner. Beside these general comments, I have several concerns:

Thank you for your time and thoughtful feedback. To address concerns regarding the 2D encoding, long scan times, and resolution, we included a multi-slice experiment to demonstrate one possible strategy for acceleration. With further enhancements, such as displacement k-space acceleration techniques, this approach can mitigate these issues and ultimately enable 3D imaging with shorter scan times and higher resolution. Additionally, we would like to note that venous territory mapping remains challenging with current methods, and our approach has the potential to significantly aid in the diagnosis of venous diseases. Finally, we conducted an additional experiment involving a third subject during a motor task to further validate our method.

Main concerns: 1. In the analysis of the motor activation, the authors seem to double-dip: first they determine T-maps where there is a change in DiSpect signal, and subsequently they select ROIs based on these T-maps to look into the DiSpect signal. Not surprisingly they find changes in DiSpect signal. This should be analyzed in a different manner.

Good point, which we need to clarify. The T-tests were used to identify regions with statistically significant differences in the mixing time signal series between baseline and task. Subsequently, we displayed the progression of the signal amplitude across mixing times for these regions to provide additional information for the reader about the temporal dynamics of the signal – rather than just “being different” which the T-test provides. We revised the manuscript to clarify the rationale (Page 10, Lines 479-481).

2. The authors claim an important role of veins in blood flow regulations. In my opinion the literature is more supportive of a passive role of the veins (following e.g. the thoracic pressure) with active control in arterioles and to a lesser extent by pericytes on capillaries. Please add references to support their claim.

Thank you for the opportunity to clarify. We do not claim that veins play an active role in neural activation or blood flow regulation but that they have significant effects on the BOLD signal in fMRI. We highlight that the venous vasculature has very complex effects on the BOLD signal. Additionally, we mention that there is a limited understanding of how changes in venous blood flow during functional activation influence the BOLD signal. While veins do not actively regulate blood flow, measuring perfusion changes in veins could provide valuable insights into the complex biophysics underlying the BOLD signal.

3. The authors do not discuss the ISMRM-abstract by Eric Wong who was the first to measure venous territories. I am aware this has never been published as full paper, but credits should be given to him and a discussion on the differences with his technique would be beneficial.

This was a mishap on our side. We are aware of it and the reference slipped through. We revised the introduction to appropriately cite and credit this study (Page 2, Lines 89-95) and highlight its distinctions from our method in the discussion. The section highlighting its distinctions from our method is (Page 15, Lines 750-765):

“Another technique, VENTI [26], was first to demonstrate the ability to map venous territories by spatially encoding tissue magnetization, similar to DiSpect, and using a phase-contrast angiographic readout to isolate intravascular signal. While VENTI can map venous territories, it does not provide any dynamic information on venous flow. In addition, the venous territories are resolved at a very low spatial resolution (2cm). Furthermore while VENTI resolves territories in the in-plane image dimensions, our approach resolves sources orthogonal to the imaging plane, enabling us to capture information across three dimensions. Finally, although the initial feasibility was demonstrated on a single subject, no further validation or analysis was performed.”

4. When introducing DiSpect the authors should already mention that the signal decays with the T1 of blood.

Certainly, we revised the manuscript to include information about the T1 decay. The following sentence (Page 3, Lines 123-124):

“Each image captures blood at longer and longer mixing times relative to the labeling ($t_{\text{mix}_1}, t_{\text{mix}_2}, t_{\text{mix}_3}, \dots$), **during which the signal also experiences T_1 decay.**”

5. Remote detection (page 3): the authors almost exclusively show cortical origins of the detected signal, i.e. also in the high SNR part of the head coil. Please reconsider this argument.

We thank the reviewer for bringing up this observation. The detected signal predominantly shows cortical origins because cortical veins drain these regions of the brain. To address this point, we conducted an additional experiment to demonstrate DiSpect’s ability to detect remote sources from deep cerebral veins. In this experiment, we placed an imaging slice lower in the brain, intersecting the straight sinus. Unlike the cortical experiments, here we performed displacement encoding along the anterior-posterior (AP) and superior-inferior (SI) directions to obtain 2D sagittal perfusion source maps.

The results of this experiment are provided in Supplementary Figure 1. When an ROI was selected on the straight sinus, we detected source signal contributions from deep venous structures, including regions in the center of the brain. Notably, sources were detected up to 15 cm away from the selected ROI, demonstrating DiSpect’s capability for remote detection. The results of this experiment are detailed in (Page 5, Lines 283-314).

6. Figure 1: From the figure it is not clear that “n” refers to a 2D spatial-encoding scheme. This could be made more clear. Similarly the repeated images, do not give an impression that these are spatially encoded maps to discern the origin of their signal. Would be great when they make this figure easier to understand.

To address this point, we modified Figure 1 to improve clarity. Specifically, we labeled all the repeated images from the different spatial encodings as s_1, s_2, \dots, s_N , hopefully making it easier to understand for the reader. Additionally, we labeled the perfusion source maps with their corresponding mixing times ($t_{\text{mix}_1}, t_{\text{mix}_2}, t_{\text{mix}_3}, \dots$), helping readers identify that these represent the same slice being imaged dynamically across time.

7. Figure 2A: it is not clear to what area the spatial saturation is applied. Also, include the SPAMM module explicitly in the diagram.

To clarify what area the spatial saturation is applied to, we updated the figure caption to explicitly state that the spatial saturation band was positioned around the imaging slice to suppress signals from static tissue. Additionally, to address another reviewer’s comment and improve the clarity of the paper, we removed the mention of SPAMM tagging from the text. Instead, we describe the tagging process as being identical to that used in the DENSE sequence. In Figure 2, we explicitly labeled the tagging block in the sequence diagram.

8. Page 13 (Discussion): please be open, how (unpractical) long the sequences did take. Please quote the scan-duration in this paragraph (near line 568).

Please also see response to reviewer 2, items (3) and (5). To address your comment, we revised the discussion section (Page 15, Lines 774-790) as follows:

“These acquisitions required scan durations of 30-45 minutes, which is impractical for routine use. However, the dynamic perfusion source maps are highly sparse and since flow is continuous they exhibit substantial temporal and spatial redundancy. In addition, the sources of blood are constrained by the venous anatomy. We already demonstrated that acquisitions can be accelerated through sequence modification such as multi-slice acquisitions, reducing the scan time to 12 minutes. Future work could further accelerate acquisitions by using partial Fourier or variable density displacement k-space sampling, and a regularized reconstruction. By combining acceleration through various different approaches, we aim to achieve significant reductions in scan time without significantly compromising SNR.”

9. Methods: this has quite some overlap with the MRM paper. Consider shortening.

Thank you for your suggestion. We revised the Methods section to make it more concise by removing redundancies and focusing only on the essential details of DENSE and DiSpect (Page 16, Lines 832-851).

Minor concerns: 10. Page 2, line 35: the venous signal change of BOLD is the main contributor to the mismatch between neuronal activation and BOLD response (so not “in addition...”).

We removed the sentence “In addition...” to avoid any confusion.

11. ASL and especially velocity-selective ASL has been used to select venous signal, also in the context of angiography (although then venous signal is often seen as something that should be avoided, making people to apply acceleration selective ASL instead).

To address your comment, we added the following sentence to the introduction (Page 2, Lines 71-76):

“Variants such as velocity-selective ASL (VSASL) can detect venous signals, but often venous signals are considered unwanted and suppressed using techniques like acceleration-selective ASL (AccASL) [22]”

12. Page 7, line 363: "...increase perfusion source map signal". A bit more help how to interpret this would be good.

We added the following sentence to that section to help with interpretation (Page 9, Lines 440-443):

"This indicates that an increased amount of blood is arriving from those source locations into the target image voxel across the corresponding mixing time."

Response to Reviewers

MR Perfusion Source Mapping Depicts Venous Territories and Reveals Perfusion Modulation during Neural Activation

Ekin Karasan Jingjia Chen Julian Maravilla Zhiyong Zhang Chunlei Liu
Michael Lustig

We would like to thank our editor and the three reviewers for their time. We are pleased that Reviewers 1 and 2 found our revisions satisfactory. We now address the two remaining comments from Reviewer 3. Below, we include Reviewer 3’s comments followed by our detailed responses in blue.

Comments of Reviewer 3:

1. Following up on my second main concern: please consider rephrasing the first sentence of the manuscript. This still seems to imply a much more active role of the venous system (although I of course recognize that the venous system is critical).

We appreciate the reviewer’s feedback. The first sentence of our manuscript currently reads: “The hemodynamics of the cerebral venous system plays a crucial role in ensuring sufficient brain perfusion to maintain cerebral function.” We believe that this statement does not imply an active regulatory role of the venous system. It is solely intended to acknowledge the critical involvement of the venous system in cerebral perfusion, consistent with the reviewer’s recognition that the venous system is critical.

2. Follow-up on point 8: Although the multislice experiment in 12 minutes is a step in the right direction, it is still very far from whole brain scanning in a feasible scantime. Therefore, this manuscript remains in the very interesting technical approach with very little clinical impact. Or can the authors include a clinical example showing the added value of their approach?

The three-slice experiment included in the manuscript was intended as a proof of concept to demonstrate the feasibility of accelerating our acquisition. These slices were strategically positioned to target the superior cerebral veins. Since three slices were sufficient to cover these veins, we did not prescribe additional slices. However, we emphasize that the current sequence is easily extensible to accommodate more slices.

For example, with a spiral readout time of approximately 15 ms, it is feasible to acquire 5–6 slices **without increasing the scan time**. Furthermore, future implementations could incorporate simultaneous multi-slice (SMS) imaging to acquire even more slices in parallel. Therefore, we believe it is technically feasible to achieve full-brain coverage within a clinically acceptable scan time.

We agree with the reviewer that demonstrating clinical utility would further strengthen the manuscript. Unfortunately, our current IRB approval does not cover patient studies, and we currently do not have access to the relevant patient population at our institution. Nevertheless, we believe that the in-vivo results and validation presented here establish a foundation for future clinical studies.